# Accuracy and clinical effectiveness of risk prediction tools for pressure injury occurrence: An umbrella review

**Bethany Hillier** [1,2], **Katie Scandrett** [1], **April Coombe** [1,2], **Tina Hernandez-Boussard** [3],
**Ewout Steyerberg** [4], **Yemisi Takwoingi** [1,2], **Vladica M. Veličković** [5,6],
**Jacqueline Dinnes** [1,2] *

1 Department of Applied Health Sciences, College of Medicine and Health, University of Birmingham, Edgbaston, Birmingham, United Kingdom, 2 NIHR Birmingham Biomedical Research Centre, University Hospitals Birmingham NHS Foundation Trust and University of Birmingham, Birmingham, United Kingdom, 3 Department of Medicine, Stanford University, Stanford, California, United States of America, 4 Department of Biomedical Data Sciences, Leiden University Medical Center, Leiden, the Netherlands, 5 Evidence Generation Department, HARTMANN GROUP, Heidenheim, Germany, 6 Institute of Public Health, Medical, Decision Making and Health Technology Assessment, UMIT, Hall, Tirol, Austria

* j.dinnes@bham.ac.uk

**Data Availability Statement:** All relevant data are within the manuscript and its Supporting Information files.

## Abstract

### Background

Pressure injuries (PIs) pose a substantial healthcare burden and incur significant costs worldwide. Several risk prediction tools to allow timely implementation of preventive measures and a subsequent reduction in healthcare system burden are available and in use. The ability of risk prediction tools to correctly identify those at high risk of PI (prognostic accuracy) and to have a clinically significant impact on patient management and outcomes (effectiveness) is not clear. We aimed to evaluate the prognostic accuracy and clinical effectiveness of risk prediction tools for PI and to identify gaps in the literature.

### Methods and findings

The umbrella review was conducted according to Cochrane guidance. Systematic reviews (SRs) evaluating the accuracy or clinical effectiveness of adult PI risk prediction tools in any clinical settings were eligible. Studies on paediatric tools, sensor-only tools, or staging/diagnosis of existing PIs were excluded. MEDLINE, Embase, CINAHL, and EPISTEMONIKOS were searched (inception to June 2024) to identify relevant SRs, as well as Google Scholar (2013 to 2024) and reference lists. Methodological quality was assessed using adapted AMSTAR-2 criteria. Results were described narratively. We identified 26 SRs meeting all eligibility criteria with 19 SRs assessing prognostic accuracy and 11 assessing clinical effectiveness of risk prediction tools for PI (4 SRs assessed both aspects). The 19 SRs of prognostic accuracy evaluated 70 tools (39 scales and 31 machine learning (ML) models), with the Braden, Norton, Waterlow, Cubbin-Jackson scales (and modifications thereof) the most evaluated tools. Meta-analyses from a focused set of included SRs showed that the scales had sensitivities and specificities ranging from 53% to 97% and 46% to 84%,

**Funding:** This work was commissioned and supported by funding from Paul Hartmann AG (Heidenheim, Germany) to the University of Birmingham (JD). This research has been delivered through the National Institute for Health and Care Research (NIHR) Birmingham Biomedical Research Centre (BRC). The funders had no role in data collection and analysis, decision to publish, or preparation of the original draft of the manuscript. All co-authors reviewed and approved the final manuscript.

**Competing interests:** VV is an employee of Paul Hartmann AG; ES and THB received consultancy fees from Paul Hartmann AG. VV, ES and THB were not involved in data curation, screening, data extraction, analysis of results or writing of the original draft. These roles were conducted independently by authors at the University of Birmingham. All other authors received no personal funding or personal compensation from Paul Hartmann AG and have declared that no competing interests exist.

**Abbreviations:** AHRQ, Agency for Healthcare Research and Quality; AMSTAR, A Measurement Tool to Assess Systematic Reviews; AUC, area under the curve; BMI, body mass index; CASP, Critical Appraisal Skills Programme; CHARMS, CHecklist for critical Appraisal and data extraction for systematic Reviews of prediction Modelling Studies; CI, confidence interval; DTA, diagnostic test accuracy; HSROC, hierarchical summary ROC; JBI, Joanna Briggs Institute; ML, machine learning; NICE, National Institute for Health and Care Excellence; PI, pressure injury; PRISMA-DTA, Preferred Reporting Items for Systematic Reviews and Meta-Analyses of Diagnostic Test Accuracy Studies; PROBAST, Prediction model Risk of Bias Assessment; QUADAS, Quality Assessment of Diagnostic Accuracy Studies; RCT, randomised controlled trial; RoB, Risk of Bias; ROC, receiver operating characteristic; RR, risk ratio; SR, systematic review; UK, United Kingdom; US, United States.

respectively. Only 2/19 (11%) SRs performed appropriate statistical synthesis and quality assessment. Two SRs assessing machine learning-based algorithms reported high prognostic accuracy estimates, but some of which were sourced from the same data within which the models were developed, leading to potentially overoptimistic results. Two randomised trials assessing the effect of PI risk assessment tools (within the full test-intervention-outcome pathway) on the incidence of PIs were identified from the 11 SRs of clinical effectiveness; both were included in a Cochrane SR and assessed as high risk of bias. Both trials found no evidence of an effect on PI incidence. Limitations included the use of the AMSTAR-2 criteria, which may have overly focused on reporting quality rather than methodological quality, compounded by the poor reporting quality of included SRs and that SRs were not excluded based on low AMSTAR-2 ratings (in order to provide a comprehensive overview). Additionally, diagnostic test accuracy principles, rather than prognostic modelling approaches were heavily relied upon, which do not account for the temporal nature of prediction.

## Conclusions

Available systematic reviews suggest a lack of high-quality evidence for the accuracy of risk prediction tools for PI and limited reliable evidence for their use leading to a reduction in incidence of PI. Further research is needed to establish the clinical effectiveness of appropriately developed and validated risk prediction tools for PI.

## Author summary

### Why was this study done?

- Pressure injuries (PIs) are injuries to and below the skin caused by prolonged pressure, especially on bony areas, and people who spend extensive periods in a bed or chair are particularly vulnerable.

- The majority of pressure injuries are preventable if appropriate preventive measures are put into place, but it is crucial to conduct risk stratification of individuals in order to appropriately allocate preventive measures.

- Numerous tools that give patients a score (or probability) to signify their risk of developing a PI exist. However, there is a lack of clarity on how accurate the risk scores are and how effective the scores are at improving patient outcomes (the clinical effectiveness) when patient management is subsequently changed for patients classified as high risk.

### What did the researchers do and find?

- We conducted an umbrella review (an overview of existing systematic reviews), identifying 26 systematic reviews which included 70 risk prediction tools.

- Of these 70 risk prediction tools, 31 were developed using machine learning (ML) methods, while the remainder were derived from statistical modelling and/or clinical expertise.

- Risk prediction tools demonstrated moderate to high accuracy, as measured by a variety of metrics. However, there were concerns regarding the quality of both the systematic reviews, and the primary studies included in these reviews, as reported by the systematic review authors.

- There were only 2 randomised controlled trials that investigated the clinical effectiveness of risk prediction tools and subsequent changes in PI management, and neither trial found that use of the tools had an impact on the incidence of PIs.

### What do these findings mean?

- While an abundance of risk prediction tools exists, it is unclear how accurate they are due to poor-quality evidence and poor reporting, so it is difficult to recommend a particular tool/tools.

- Even if the tools are shown to be accurate, they are not useful unless they lead to improvement in patient outcomes. There is very limited evidence to determine whether the tools are clinically effective and the evidence that does exist suggests that the tools did not lead to improved patient outcomes.

- More research into the clinical effectiveness of appropriately developed and evaluated tools, when they are adopted within the clinical pathway, is needed.

- The main limitations of this study are the use of the quality assessment tool, the poor reporting quality of the included reviews, and the reliance on particular statistical methods to assess the accuracy of the risk prediction tools, which do not consider the time interval between use of the tool (prediction) and the onset of a PI occurring (outcome).

### Introduction

Pressure injuries (PI), also known as pressure ulcers or decubitus ulcers, have an estimated global prevalence of 12.8% among hospitalised adults [1] and place a significant burden on healthcare systems (estimated at $26.8 billion per year in the United States (US) alone [2]). PIs are most common in individuals with reduced mobility, limited sensation, poor circulation, or compromised skin integrity, and can affect those in community settings and long-term care as well as hospital settings. Effective prevention of PI requires multicomponent preventive strategies such as mattresses, overlays and other support systems, nutritional supplementation, repositioning, dressings, creams, lotions, and cleansers [3,4]. Health economic models have suggested that providing baseline preventive interventions for all with daily risk assessments is more cost-effective than either a less standardised prevention protocol or a targeted risk-stratified prevention strategy [5]. Nevertheless, the stratification of patients by risk could further improve outcomes by allowing timely and targeted implementation of additional or greater intensity preventive measures in those most at risk, to reduce harm and consequently burden to healthcare systems [6].

Numerous clinical assessment scales and statistical risk prediction models for assessing the risk of PI are available. However, the methodology underlying their development is not always explicit, with scales in routine clinical usage apparently based on epidemiological evidence and clinical judgement about predictors that may not meet accepted principles for the development and reporting of risk prediction models [7]. The Braden [8,9], Norton [10], and Waterlow [11] scales are recommended by the National Institute for Health and Care Excellence (NICE) guidelines [12] in the United Kingdom (UK) and referenced in international guidelines for PI prevention [13]. In some hospitals and long-term care settings in the US, healthcare professionals must conduct mandatory risk assessments for PI for all patients for the purposes of risk stratification and clinical triage. The Braden scale, developed in 1987 using a sample of 102 older hospital patients in the US includes sensory perception, moisture, activity, mobility, nutrition, friction, and shear as predictors [8,9]. The Norton scale, based on a sample of 250 older hospital patients in the UK and published in 1962, includes physical condition, mental status, activity, mobility, and continence domains [10]. The Waterlow scale was published in 1985 for use by Waterlow's nursing students in the UK [14], and assesses body mass index (BMI), assessment of the skin, sex, age, malnutrition, incontinence, mobility, tissue malnutrition, neurological deficits, major surgery or trauma, and medication [11].

There is a considerable body of evidence evaluating the psychometric properties and clinical utility of available risk prediction tools, much of which has been synthesised in systematic reviews and meta-analyses [7]. However, there is an apparent lack of reporting of now standard methods for development and validation of risk prediction tools. Clinical utility includes both prognostic accuracy and clinical effectiveness. Prognostic accuracy is estimated by applying a numeric threshold above (or below) which there is a greater risk of PI, with study results presented using accuracy metrics such as sensitivity, specificity or the area under the receiver operating characteristic (ROC) curve [15]. Resulting accuracy is driven not only by the nominated threshold for defining participants as at low or high risk for PI but by other study factors including population and setting [16]. Clinical effectiveness, or the ability of a tool to ultimately impact on health outcomes such as the incidence or severity of PI, is related both to the accuracy of the tool (or its ability to correctly identify those most likely to develop PI), to the uptake and implementation of the tool in practice and to the consequential changes in PI management based on tool predictions. Demonstrating a change in health outcomes as a result of use of a risk prediction tool is vital to encourage implementation [17].

Using an umbrella review approach, we aimed to provide a comprehensive overview of available systematic reviews that consider the prognostic accuracy and clinical effectiveness of PI risk prediction tools.

## Methods

### Protocol registration and reporting of findings

We followed Cochrane guidance for conducting umbrella reviews [18], and "Preferred Reporting Items for Systematic Reviews and Meta-Analyses of Diagnostic Test Accuracy Studies" (PRISMA-DTA) reporting guidelines [19] (see Appendix A in S1 Supporting Information). The protocol was registered on Open Science Framework (https://osf.io/tepyk).

### Literature search

Electronic searches of MEDLINE, Embase via Ovid, and CINAHL Plus EBSCO from inception to June 2024 were developed and conducted by an experienced information specialist (AC), employing well-established systematic review and prognostic search filters [20–22], combined with appropriate keywords related to PIs. Simplified supplementary searches in

EPISTEMONIKOS and Google Scholar were also undertaken, with the latter covering the years 2013 to June 2024 (see Appendix B in S1 Supporting Information for further details). Screening of search results and full texts were conducted independently and in duplicate by any 2 from a group of 4 reviewers (BH, JD, YT, and KS), with arbitration by a third reviewer where necessary (any one of the 4 reviewers not involved in the independent screening).

## Eligibility criteria for this umbrella review

Published English-language systematic reviews of risk prediction tools developed for adult patients at risk of PI in any setting were included. We understand the term "adult" to refer to individuals aged 18 and over, but accepted individual study definitions of adult and also included studies in which "adult" was not defined. Studies focused on tools developed for paediatric populations, as defined by tool developers, were excluded. Clinical risk assessment scales and models developed using statistical or machine learning (ML) methods were eligible (models exclusively using pressure sensor data were not considered). Risk prediction tools could be applied by any healthcare professional using any threshold for classifying patients as high or low risk and using any PI classification system [13,23–25] as a reference standard. For prognostic accuracy, we required accuracy metrics, such as sensitivity and specificity, to be presented but did not require full 2 × 2 classification tables to be reported. Reviews on diagnosing or staging suspected or existing PIs were excluded.

To be considered "systematic," reviews were required to report a thorough search of at least 2 electronic databases and at least one other indication of systematic methods (e.g., explicit eligibility criteria, formal quality assessment of included studies, adequate data presentation for reproducibility of results, or review stages (e.g., search screening) conducted independently in duplicate).

## Data extraction and quality assessment

Data extraction forms (Appendix C in S1 Supporting Information) were informed by the CHARMS checklist (CHecklist for critical Appraisal and data extraction for systematic Reviews of prediction Modelling Studies) and Cochrane Prognosis group template [26,27]. Data extraction items included review characteristics, number of studies and participants, study quality, and results.

The methodological quality of included systematic reviews was assessed using an adapted version of AMSTAR-2 (A Measurement Tool to Assess Systematic Reviews) [28]. For example, for reviews evaluating the prognostic accuracy of risk prediction tools we assessed eligibility criteria using the PIRT framework (Population, Index test, Reference standard, Target condition) [29] and POII framework (Population, Outcome to be predicted, Intended use of model, Intended moment in time) [30] and required methodological quality assessment to be conducted using validated and appropriate tools such as QUADAS (Quality Assessment of Diagnostic Accuracy Studies) [31], QUADAS-2 [32], or PROBAST (Prediction model Risk of Bias Assessment) [33]. We omitted the AMSTAR-2 item relating to publication bias (Item 15) because of the lack of empirical evidence for the effect of publication bias on test accuracy estimates and limitations in statistical methods for identifying publication bias [19,34]. Our adapted AMSTAR-2 contains 6 critical items, and limitations in any of these items reduces the overall validity of a review [28]. Full details can be found in Appendix D in S1 Supporting Information. Quality assessment and data extraction were conducted by one reviewer and checked by a second (BH, JD, KS), with disagreements resolved by consensus.

## Synthesis methods

Reviews about prognostic accuracy and clinical effectiveness of risk prediction tools were considered separately. Review methods and results were tabulated, and a narrative synthesis

provided. Prognostic accuracy results from reviews including a statistical synthesis were tabulated according to risk prediction tool.

Considerable overlap in risk prediction tools and included primary studies was noted between reviews. For risk prediction tools that were included in multiple meta-analyses, we focused our synthesis on the review(s) with the most recent search date or most comprehensive (based on number of included studies) and most robust estimate of prognostic accuracy (judged according to the appropriateness of the meta-analytic method used, e.g., use of recommended hierarchical approaches for test accuracy data [35]). The prognostic accuracy of risk prediction tools that were included in 3 or fewer reviews was reported only if an appropriate method of statistical synthesis [18] was used.

For clinical effectiveness results, reviews with the most recent search date or most comprehensive overview of available studies that assessed PI incidence outcomes and that at least partially met more of the AMSTAR-2 criteria [28] were prioritised for narrative synthesis.

## Results

### Characteristics of included reviews

A total of 118 records were selected for full-text assessment from 7,200 unique records. We could obtain the full text of 111 publications, of which 26 reviews met all eligibility criteria (Fig 1), 19 reported accuracy data [36–54] and 11 reported clinical effectiveness data [38,42,43,49,55–61] (4 reported both accuracy and effectiveness data [38,42,43,49]). Table 1 and Fig 2 provide an overview of the characteristics, methods, and methodological quality of all 26 reviews (see Appendix E in S1 Supporting Information for full details).

Reviews were published between 2006 and 2024. Over half (15/26, 58%) restricted inclusion to adult populations (Table 1), 2 (2/26, 8%) included any age group, and 9 (9/26, 35%) did not report any age restrictions. Six reviews (6/26, 23%) only included study populations with no PI at baseline. Acute care was the most frequent setting across both review questions (7/19 (37%) accuracy reviews and 3/11 (27%) effectiveness reviews). Quality assessment tools included QUADAS-2 ($n = 8$) or QUADAS ($n = 2$) in more than half of reviews of accuracy (10/19, 53%). One review [47] utilised and reported PROBAST assessments for risk of bias. Another review [48] reported using QUADAS-2 and PROBAST tools in their methods, but only reported QUADAS-2 results.

Reviews of accuracy either included studies evaluating any tool (5/19, 26%) or prespecified tools (10/19, 53%); 2 [47,48] included only ML-based prediction models, while the remaining 2 [49,50] did not specify the tools to be included. A total of 70 risk prediction tools were reported across the reviews (from one [37,40,41,46,51,52] to 28 [39] tools included per review), including 31 ML models. Only 2 reviews (2/19, 11%) reported eligibility criteria related to the development or validation of the risk prediction tools. One [43] excluded evaluation studies that used the same data that was used to develop the tool and the other [38] included only "validated risk assessment instruments" with no further definition (yet included studies reporting original tool development).

The majority (15/19, 79%) of accuracy reviews conducted a meta-analysis, but only 2 utilised currently recommended hierarchical approaches for the meta-analysis of test accuracy data [41,53]. Eight reviews conducted univariate meta-analysis of individual accuracy measures (e.g., sensitivity and specificity separately, or area under the curve (AUC) [50], risk ratios (RRs) [39], or odds ratio [43]) and 5 did not clearly report the type of analysis approach used.

Of the 11 systematic reviews evaluating clinical effectiveness, 2 only considered the reliability of risk assessment scales [49,58], 1 considered reliability and other "psychometric" properties [42], and 8 considered effects on patient outcomes (one of which also considered tool

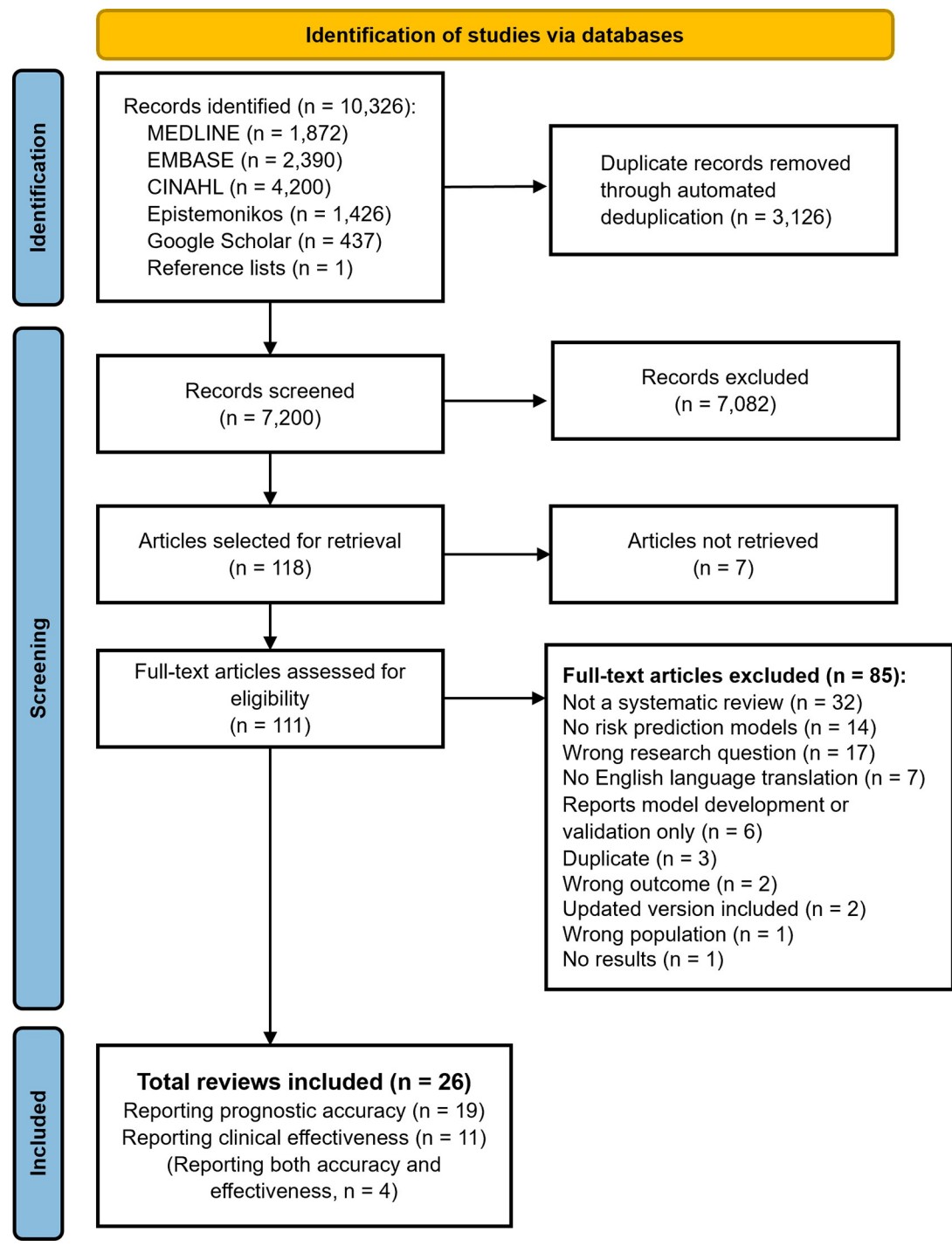

**Fig 1. PRISMA flowchart: Identification, screening, and selection process.** List of full-text articles excluded, with reasons, is given in Appendix E in S1 Supporting Information.

reliability [55]). More than half of reviews (6/11, 55%) compared use of PI risk assessment scales to clinical judgement alone or "standard care." The number of included studies ranged from 1 [56] to 20 [60], with sample sizes ranging from 1 (1 subject and 110 raters, in an inter-rater reliability study [62]) to 4,137 patients. Reported outcomes included the incidence of PIs

**Table 1.** Summary of included systematic review characteristics.

| Review characteristic | Reviews on prognostic accuracy of risk prediction tools (N = 19) | Reviews on clinical effectiveness of risk prediction tools (N = 11) | All included reviews (N = 26) |
|---|---|---|---|
| Median (range) year of publication | 2016 (2006–2024) | 2015 (2006–2024) | 2017 (2006–2024) |
| **Eligibility criteria** | | | |
| **Participants** | | | |
| Adults only | 11 (58) [A] | 6 (55) | 15 (58) [A] |
| Review states "Any age" | 1 (5) | 1 (9) | 2 (8) |
| No age restriction reported | 7 (37) | 4 (36) | 9 (35) |
| **Presence of PI at baseline** | | | |
| Excluded those with PI at baseline[B] | 5 (26) | 2 (18) | 6 (23) |
| NS | 14 (74) | 8 (73) | 20 (77) |
| **Setting** | | | |
| Any healthcare setting | 1 (5) | 1 (9) | 2 (8) |
| Hospital | 3 (16) | 0 (0) | 3 (12) |
| Acute care (incl. surgical and ICU) | 7 (37) | 3 (27) | 8 (31) |
| Hospital or acute care | 0 (0) | 2 (18) | 2 (8) |
| Long-term care | 2 (11) | 0 (0) | 2 (8) |
| Long-term, acute or community settings | 1 (5) | 1 (9) | 1 (4) |
| NS | 5 (26) | 4 (36) | 8 (31) |
| **Risk assessment tools** | | | |
| Any prediction tool or scale | 5 (26) | 6 (55) | 9 (35) |
| Specified clinical scale(s) | 10 (53) | 3 (27) | 12 (46) |
| ML-based prediction models | 2 (11) | 0 (0) | 2 (8) |
| PI prevention strategies | 0 (0) | 1 (9) | 1 (4) |
| NS | 2 (11) | 1 (9) | 2 (8) |
| **PI classification system** | | | |
| Any PI classification system | 1 (5) | 0 (0) | 1 (4) |
| Accepted standard classifications | 1 (5) | 1 (9) | 2 (8) |
| Several specified classification systems (NPUAP, EPUAP, AHCPR, or TDCPS) | 3 (16) | 1 (9) | 3 (12) |
| PI stage predefined [63]/defined by study authors | 1 (5) | 0 (0) | 1 (4) |
| NS | 13 (68) | 9 (82) | 19 (73) |
| **Source of data** | | | |
| Prospective only | 4 (21) | 2 (18) | 4.5 (17) [C] |
| Prospective or retrospective | 1 (5) | 2 (18) | 2.5 (10) [C] |
| NS | 14 (74) | 7 (64) | 19 (73) |
| **Study design restrictions** | | | |
| Yes | 9 (47) | 7 (64) | 14 (54) |
| No | 2 (11) | 2 (18) | 3 (12) |
| NS | 8 (42) | 2 (18) | 9 (35) |
| **Phase of development/evaluation of tools** | | | |
| Development studies (with internal/external validation NS) | 1 (5) | N/A | N/A |
| External evaluations only | 2 (11) | N/A | N/A |
| Validation studies (internal or external NS) | 1 (5) | N/A | N/A |
| NS | 15 (79) | N/A | N/A |
| **Review methods** | | | |
| **Median (range) no. sources[D] searched** | 6 (2–14) | 5 (3–14) | 6 (2–14) |
| **Publication restrictions:** | | | |

(*Continued*)

**Table 1.** (Continued)

| Review characteristic | Reviews on prognostic accuracy of risk prediction tools (N = 19) | Reviews on clinical effectiveness of risk prediction tools (N = 11) | All included reviews (N = 26) |
|---|---|---|---|
| **Median (range) year of publication** | **2016 (2006–2024)** | **2015 (2006–2024)** | **2017 (2006–2024)** |
| **End date (year)** | | | |
| 2000–2009 | 1 (5) | 3 (27) | 3 (12) |
| 2010–2019 | 12 (63) | 6 (55) | 16 (62) |
| 2020–2023 | 6 (32) | 2 (18) | 7 (27) |
| **Language** | | | |
| English only | 7 (37) | 7 (64) | 10 (38) |
| 2 languages | 1 (5) | 2 (18) | 3 (12) |
| >2 languages | 2 (11) | 2 (18) | 3 (12) |
| No restrictions | 3 (16) | 1 (9) | 4 (15) |
| NS | 6 (32) | 0 (0) | 6 (23) |
| **Quality assessment tool** [E] | | | |
| PROBAST | 1 (5) [F] | N/A | 1 (4) [F] |
| QUADAS | 2 (11) | N/A | 2 (8) |
| QUADAS-2 | 8 (42) | N/A | 8 (31) |
| JBI tools | 1 (5) | 2 (18) | 3 (12) |
| CASP | 2 (11) | 1 (9) | 2 (8) |
| Cochrane RoB tool | 0 (0) | 1 (9) | 1 (4) |
| Other | 2 (5) | 5 (45) | 6 (23) |
| None | 3 (16) | 2 (18) | 4 (15) |
| **Meta-analysis included** | 15 (79) | 0 (0) | 15 (58) |
| **Method of meta-analysis (% of reviews incl. meta-analysis)** | | | |
| Univariate RE/FE model (depending on heterogeneity assessment) | 2 (13) [G] | N/A | N/A |
| Univariate RE model | 6 (40) [G] | N/A | N/A |
| Hierarchical model (for DTA studies) | 2 (13) | N/A | N/A |
| Unclear/NS | 5 (33) [G] | N/A | N/A |
| **Volume of evidence** | | | |
| **Median (range) no. studies** | 18 (2–70) | 5 (1–20) | 15 (1–70) |
| **Median (range) no. participants** | 13,464 (609–408,504) | 1,951 (528–5,052) | 7,684 (528–408,504) |
| **Median (range) no. tools** | 3 (1–28) | 3 (1–9) | 3 (1–28) |

Figures are number (%) of reviews, unless otherwise specified.

[A] One review [45] restricted to aged >60 years.

[B] "baseline" refers to beginning of study or hospital admission depending on included reviews.

[C] One review [38] states either prospective or retrospective data eligible for Research Question 1, but prospective only for Research Question 2, hence 0.5 added to each category.

[D] Including databases, bibliographies or registries.

[E] Reviews may fall into multiple categories; therefore, total number within domain not necessarily equal to N (100%).

[F] Another review [48] reported use of PROBAST in methods, but did not present any PROBAST results, therefore not included.

[G] One review conducts univariate meta-analysis for single estimate, e.g., AUC [50], RR [39], or OR [43].

AHCPR, Agency for Health Care Policy and Research; AUC, area under the curve; CASP, Critical Appraisal Skills Programme; DTA, diagnostic test accuracy; EPUAP, European Pressure Ulcer Advisory Panel; FE, fixed effects; ICU, intensive care unit; JBI, Joanna Briggs Institute; ML, machine learning; NPUAP, National Pressure Ulcer Advisory Panel; NS, not stated; OR, odds ratio; PI, pressure injury; PROBAST, Prediction model Risk of Bias Assessment; QUADAS (2), Quality Assessment of Diagnostic Accuracy Studies (Version 2); RE, random effects; RoB, Risk of Bias; RR, risk ratio; TDCPS, Torrance Developmental Classification of Pressure Sore.

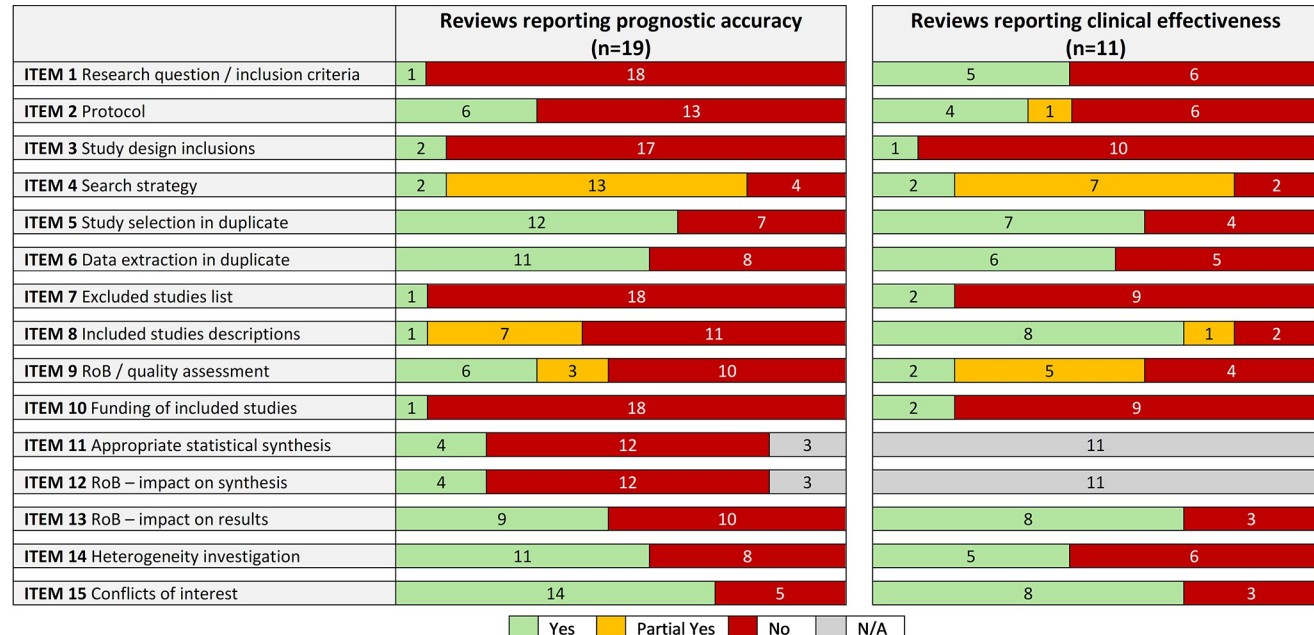

**Fig 2. Summary of AMSTAR-2 assessment results.** Item 1 –Adequate research question/inclusion criteria?; Item 2 –Protocol and justifications for deviations?; Item 3 –Reasons for study design inclusions?; Item 4 –Comprehensive search strategy?; Item 5 –Study selection in duplicate?; Item 6 –Data extraction in duplicate?; Item 7 –Excluded studies list (with justifications)?; Item 8 –Included studies description adequate?; Item 9 –Assessment of RoB/ quality satisfactory?; Item 10 –Studies' sources of funding reported?; Item 11 –Appropriate statistical synthesis method?; Item 12 –Assessment of impact of RoB on synthesised results?; Item 13 –Assessment of impact of RoB on review results?; Item 14 –Discussion/investigation of heterogeneity?; Item 15 – Conflicts of interest reported?; N/A–Not Applicable; RoB–Risk of Bias. Further details on AMSTAR items are given in Appendix D in S1 Supporting Information, and results per review are given in Appendix E in S1 Supporting Information.

(7/11, 64%), preventive interventions prescribed (5/11, 45%) and inter-rater reliability (4/11, 36%), internal consistency, measurement error and convergent validity (1/11, 9%) (latter 4 properties reported in Appendix E in S1 Supporting Information). One review [61] used the Cochrane risk of bias (RoB) tool for quality assessment of included studies, 2 reviews [59,60] used Joanna Briggs Institute (JBI) tools and another review [43] used the Critical Appraisal Skills Programme (CASP) checklist. Due to heterogeneity in study design, risk prediction tools, and outcomes evaluated, none of the included reviews provided any form of statistical synthesis of study results.

## Methodological quality of included reviews

The quality of included reviews was generally poor (Table 2; Appendix E in S1 Supporting Information). The AMSTAR-2 items that were most consistently met (yes or partial yes) were: comprehensiveness of the search (21/26, 81%), study selection independently in duplicate (17/ 26, 65%), data extraction in independently in duplicate (15/26, 58%), and conflicts of interest reported (20/26, 77%).

Six (6/19, 32%) accuracy reviews [36,40,41,47,48,53] and 2 (2/11, 18%) effectiveness reviews used an appropriate method of quality assessment of included studies (i.e., QUADAS or QUADAS-2 dependent on publication year, or PROBAST for accuracy and the Cochrane tool for assessing risk of bias [64] and criteria consistent with AHRQ (Agency for Healthcare Research and Quality) Methods Guide for Effectiveness and Comparative Effectiveness Reviews [65] for effectiveness reviews) and also presented judgements per study. Five reviews either reported quality assessment results per study (*n* = 4 [42,58–60]) or were considered to use an appropriate quality assessment tool (*n* = 1 [43]) (AMSTAR-2 criterion partially met).

**Table 2. Findings related to prognostic accuracy, by model: Characteristics and quality of studies included within reviews.**

| Review author (publication year) | Tool(s) evaluated n studies; N participants | Brief description of included studies | Brief description of included study quality | Method of meta-analysis |
|---|---|---|---|---|
| Huang [41] (2021) | **Braden** [8,9] n = 60; N = 49,326 | Setting: hospital (n = 45; includes 22 in ICU or other acute units), LTCF (n = 15) Sample size: 25 to 10,098 Mean age: range 31.7 ± 10.9 to 84.6 ± 7.9 Design: 47 prospective, 13 retrospective Braden cut-off (out of 23): range ≤10 to ≤20 Development study not included. | QUADAS-2: Studies performed worst overall in the patient selection domain, with low RoB in only 11/60 (18%) and high concern for applicability in 16/60 (27%). The index test domain also revealed some issues, with about one-third of studies having high RoB. Studies performed well in the remaining domains. | Bivariate meta-analysis, SROC constructed, and subgroup/stratified analyses to explore heterogeneity. |
| Chen [36] (2023) | **Cubbin and Jackson** [69] n = 9; N = 7,684 | Study designs included prospective studies (n = 2), a prospective and cross-sectional study (n = 1), retrospective studies (n = 3), an observational study (n = 1), a predictive correlational study (n = 1) and a longitudinal study (n = 1). Studies were conducted in South Korea (n = 3), the US (n = 2), Turkey (n = 2), China (n = 1), and Portugal (n = 1). Studies targeted research participants >18 years old (n = 6), >16 years old (n = 1), ≥21 years old (n = 1), or did not set an age limit (n = 1). All studies restricted to patients without PIs on ICU admission. | Authors state that *"none of [the 9 included studies] has a high risk of bias in any field,"* despite indicating some as "high risk" in the table of QUADAS-2 results. From table of QUADAS-2 results: 5/9 (56%) studies are at high RoB, 3/9 (33%) are at unclear RoB, and 1/9 (11%) is at low RoB. | Method unclear, but synthesised across different thresholds of Cubbin and Jackson; SROC constructed. |
| Zhang [53] (2021) | **4 tools evaluated in meta-analyses** | Studies not described according to prediction tool. All prospective. | Studies not described according to prediction tool QUADAS-2: Overall judgement was "not so satisfactory." | HSROC model for >3 studies, or univariate fixed- or random-effects models if ≤3 studies; meta-regression heterogeneity investigation. |
| | **Braden** [8,9] n = 18; N = 11,167 | Cut-offs used range from 10.5 to 20. Development study not included. | | |
| | **Cubbin and Jackson** [69] n = 6; N = 800 | Cut-offs used range from 24 to 34. Development study not included. | | |
| | **EVARUCI** [70] n = 3; N = 3,063 | Cut-offs: >11, >11.5, NS Development study not included. | | "Inconsistency" (I² statistic) of studies was found to be 0%, therefore univariate fixed-effects models used. |
| | **Waterlow** [11] n = 4; N = 1,000 | Cut-offs: 12, 16, <25, 20.5 Development study not included. | | |
| Park [45] (2016b) | **3 tools evaluated** | Studies not described according to prediction tool Cut-offs selected "by following the one which the study researcher(s) indicated to be the most effective" | | DTA meta-analysis (random effects) using MetaDiSc; no further details |
| | **Braden** [8,9] n = 25; N = 10,547 | Cut-offs: 13 (n = 2); 16 (n = 8); 17 (n = 2); 18 (n = 9); 19 (n = 3); 20 (n = 1) Development study not included. | | |
| | **Norton** [10] n = 5; N = 2,408 | Cut-offs: 14 (n = 2); 16 (n = 3) Development study not included. | | |
| | **Waterlow** [11] n = 5; N = 1,406 | Cut-offs: 15 (n = 1); 16 (n = 2); 17 (n = 1); NS (n = 1) Development study not included. | | |

*(Continued)*

**Table 2.** (Continued)

| Review author (publication year) | Tool(s) evaluated *n* studies; *N* participants | Brief description of included studies | Brief description of included study quality | Method of meta-analysis |
|---|---|---|---|---|
| Park [44] (2016a) | **5 tools evaluated** | Described below according to prediction tool | QUADAS-2: Studies not described according to prediction tool "None had 'high risk'" | DTA meta-analysis (random-effects) using MetaDiSc; Cochrane Handbook (2010) [34] and Walter 2002 [80] cited |
| | **Braden– modified by Song and Choi** [67] *n* = 4; *N* = 688 | Prospective (4/4), recruiting patients with no PI at baseline (hospital ward (*n* = 2) or ICU (*n* = 3); mean age in the 50s (*n* = 2), 60 s (*n* = 2)). Classification used: AHCPR (*n* = 3), Bergstrom (*n* = 1). Braden scale cut-off used: <21 (*n* = 1), <23 (*n* = 1), <24 (*n* = 2) Development study not included. | | |
| | **Braden– modified by Pang and Wong** [68] *n* = 2; *N* = 626 | Prospective (2/2), recruiting patients with no PI at baseline (OS ward (*n* = 1) or NS (*n* = 1); mean age 79.4 and 54.1). Classification used: NPUAP (*n* = 2) Braden scale cut-off used: <19 (*n* = 1), <14 (*n* = 1). Development study not included. | | |
| | **Cubbin and Jackson** [69] n = 4; N = 662 | Prospective (4/4); ICU patients for all studies (1 in surgical ICU), with no PI at baseline (*n* = 3); mean age in the 50s (*n* = 2), 60s (*n* = 2). Classification used: AHCPR (*n* = 2), NPUAP (*n* = 1), Lowthian (*n* = 1). C&J scale cut-off used: <24 (*n* = 2), <26 (*n* = 1), <28 (*n* = 1). Development study not included. | | |
| | **Norton** [10] *n* = 7; *N* = 2,899 | Prospective (6/7); inpatients with no PI at baseline (1 LTC, 2 "hospital," 1 ICU, 1 ICU and wards); mean age in the 50s (*n* = 1), 60s (*n* = 3), or 80s (*n* = 1), or NS (*n* = 2). Classification used: AHCPR (*n* = 3), NPUAP (*n* = 2), EPUAP (*n* = 1), TDCPS (*n* = 1). Norton scale cut-off used: <14 (*n* = 2, but reported as 3 in paper), <15 (*n* = 2), <16 (*n* = 3). Development study not included. | | |
| | **Waterlow** [11] *n* = 6; *N* = 1,268 | Prospective (6/6); all male* inpatients aged over 60 on average with no PI at baseline (3 included ICU patients). Classification used: AHCPR (*n* = 2), NPUAP (*n* = 2), EPUAP (*n* = 1), TDCPS (*n* = 1). Waterlow scale cut-off used: <9 (*n* = 1), <15 (*n* = 1), <16 (*n* = 2), <17 (*n* = 1), NS (*n* = 1). Development study not included. | | |

*(Continued)*

**Table 2.** (Continued)

| Review author (publication year) | Tool(s) evaluated *n* studies; *N* participants | Brief description of included studies | Brief description of included study quality | Method of meta-analysis |
|---|---|---|---|---|
| Pei [47] (2023) | **Various ML models** *n* = 14; *N* = 408,504 | Studies published from 2012–2022; studies conducted in China (*n* = 4), Taiwan (*n* = 3), USA (*n* = 6), Germany (*n* = 1), Japan (*n* = 1), Australia (*n* = 1), Spain (*n* = 1), and Czech Republic (*n* = 1); 15 studies utilised retrospective data, while 3 used prospective data; studies focused on adult ICU inpatients (*n* = 5), hospitalised patients (*n* = 8), adult hospitalised patients awaiting surgery (*n* = 3), cancer patients (*n* = 1), and end-of-life adult inpatients (*n* = 1); sample size range 168–149,006; number of patients with PI range 8–4,663. Validation methods: both sample splitting and k-fold cross-validation (*n* = 10), sample splitting only (*n* = 4), k-fold cross-validation only (*n* = 1). External verification conducted (*n* = 1). Validation methods not reported (*n* = 2). Handling of missing data not disclosed (*n* = 7). | PROBAST: Overall, 16/18 (89%) papers were at high RoB, 1/18 (6%) was at unclear RoB and only 1/18 (6%) was at low RoB; 14/18 (78%) studies were at high RoB in the analysis domain. The most common factors contributing to the high risk of bias in the analysis domain included an inadequate number of events per candidate predictor, poor handling of missing data, and failure to deal with overfitting. | Meta-analysis conducted for the 14 studies that presented 2 × 2 information; DTA meta-analysis (random-effects) using MetaDiSc (no further details); SROC constructed; meta-regression heterogeneity investigation (only *p*-values reported). |
| Qu [48] (2022) | Models by ML algorithm type | Characteristics only reported overall, not by algorithm type. Conducted in: hospital patients (*n* = 13); surgical patients (*n* = 3), ICU (*n* = 5), CVD patients (*n* = 2), cancer patients (*n* = 1), LTC (*n* = 1). Unclear whether development, internal validation, or external validation studies included. | | RevMan (Moses and Littenberg method [81,82]) for analysis of quantitative data (SROC plot presented) and Bayesian DTA-NMA. |
| | **Decision Tree** *n* = 14; *N* = 118,292 | | 2/14 (14%) high RoB; 10/14 (71%) low RoB; 2/14 (14%) unclear RoB. | |
| | **Logistic Regression** *n* = 14; *N* = 195,927 | | 4/14 (29%) high RoB; 9/14 (64%) low RoB; 1/14 (7%) unclear RoB. | |
| | **Neural Network** *n* = 9; *N* = 97,815 | | 1/9 (11%) high RoB; 7/9 (78%) low RoB; 1/9 (11%) unclear RoB. | |
| | **Random Forest** *n* = 7; *N* = 161,334 | | 1/7 (14%) high RoB; 6/7 (86%) low RoB. | |
| | **Support Vector Machine** *n* = 9; *N* = 152,068 | | 1/9 (11%) high RoB; 8/9 (89%) low RoB. | |

\* As reported in review's text. However, the table reports a mixture of female and male participants for all studies, with a mean female proportion of 50.73%.
AHCPR, Agency for Health Care Policy and Research; CI, confidence interval; CVD, cardiovascular disease; DTA, diagnostic test accuracy; EPUAP, European Pressure Ulcer Advisory Panel; (H)SROC, (hierarchical) summary receiver operating characteristic curve; ICU, intensive care unit; LTC(F), long-term care (facility); ML, machine learning; N, number of participants; n, number of studies; NMA, network meta-analysis; NS, not stated; NPUAP, National Pressure Ulcer Advisory Panel; PI, pressure injury; PPPU, Panel for the Prediction and Prevention of Pressure Ulcers; PROBAST, Prediction model Risk of Bias Assessment; QUADAS, Quality Assessment of Diagnostic Accuracy Studies; RoB, risk of bias; TDCPS, Torrance Developmental Classification of Pressure Sore; USA, United States of America.

Of the accuracy reviews that included a statistical synthesis, 25% (4/16) [39,41,50,53] used an appropriate meta-analytic method and investigated sources of heterogeneity. Two reviews [41,53] used recommended hierarchical approaches to meta-analysis of test accuracy data (the bivariate model [41] and hierarchical summary ROC (HSROC) model [53]) and 2 reviews calculated summary estimates of individual measures, using random effects meta-analyses (AUC [50] or RR [66]).

Compared to the reviews of accuracy, reviews of effectiveness more commonly provided adequate descriptions of primary studies (8/11, 73% versus 1/19, 5%) and adequately defined their inclusion criteria (5/11, 45% versus 1/19, 5%) (Fig 2). No other major differences across review questions were noted.

## Results from reviews evaluating the prognostic accuracy of risk prediction tools

Seven of 19 accuracy reviews were prioritised for narrative synthesis (Tables 2 and 3) and are reported below according to risk prediction tool. Five of the 7 reviews did not include development study estimates within their meta-analyses, one review of ML models did not report this information [48] and one [47] restricted inclusion to studies reporting model development studies. The latter review was the only one to consider the effect of study quality in their statistical syntheses.

**Braden and modified Braden scales.** The most recent and largest review [41] of the Braden scale (60 studies, including 49,326 patients), which used hierarchical bivariate meta-analysis, reported an overall summary sensitivity of 0.78 (95% confidence interval (CI) [0.74, 0.82]; 15,241 patients) and specificity of 0.72 (95% CI [0.66, 0.78]; 34,085 patients) across all reported thresholds (range ≤10 to ≤20). Summary sensitivities and specificities ranged from 0.79 (95% CI [0.76, 0.82]) and 0.66 (95% CI [0.55, 0.75]) at the lowest cut-offs for identification of high-risk patients (≤15 in 15 studies) to 0.82 (95% CI [0.73, 0.89]) and 0.70 (95% CI [0.62, 0.77]) using a cut-off of 18 (15 studies), respectively. Heterogeneity investigations suggested higher accuracy for predicting PI risk in patients with a mean age of 60 years or less, in hospitalised patients (compared to long-term care facility residents) and in Caucasian populations (compared to Asian populations) [41]. The review noted a high risk of bias for the "index test" section of the QUADAS-2 assessment in approximately a third of included studies, but failed to provide further details.

Two modified versions of the Braden scale [67,68] were included in another review [44]. Summary sensitivities were 0.97 (95% CI [0.92, 0.99]; 125 patients from 4 studies) [67] and 0.89 (95% CI [0.71, 0.98]; 27 patients from 2 studies) [68], and summary specificities were 0.70 (95% CI [0.66, 0.73]; 563 patients) [67] and 0.71 (95% CI [0.67, 0.75]; 599 patients) [68]. The review was rated critically low on the AMSTAR-2 assessment, with only 1/15 (13%) criteria fulfilled. QUADAS-2 was reportedly used but results not reported in any detail, other than to indicate that none of the included studies were considered at high risk of bias.

**Cubbin and Jackson scale.** The most recent and comprehensive review [36] of the Cubbin and Jackson scale (9 studies, including 7,684 patients) reported summary sensitivity of 0.81 (95% CI [0.51, 0.95]; 1,558 patients) and specificity of 0.76 (95% CI [0.58, 0.88]; 6,126 patients). However, this review scored critically low on AMSTAR-2 (3/15, 20%, criteria fulfilled), with authors utilising inappropriate methods for statistical synthesis, not investigating causes of heterogeneity and poor reporting of results throughout. Their meta-analysis approach was also not clearly reported, but it appears that univariate meta-analyses were conducted separately for sensitivity and specificity, across studies with different Cubbin and Jackson thresholds.

Zhang and colleagues [53] included 6 studies evaluating the original Cubbin and Jackson scale [69] (800 patients). Summary sensitivity and specificity were both reported as 0.84 (95% CIs [0.59, 0.95] and [0.66, 0.93], respectively) [53] suggesting that this represents the point on the HSROC curve where sensitivity equals specificity, particularly as reported thresholds ranged from 24 to 34. The review authors concluded that although the accuracy of the Cubbin and Jackson scale was higher than the EVARUCI scale and the Braden scale, low quality of evidence and significant heterogeneity limit the strength of conclusions that can be drawn.

**Table 3. Summary estimates of accuracy parameters (main results from statistical syntheses), by prediction tool.**

| Review author (publication year) | n studies; N participants | Sensitivity (95% CI) (N = no. participants with PI) | Specificity (95% CI) (N = no. participants without PI) | Likelihood ratios (95% CI) | DOR (95% CI) | AUROC (95% CI) |
|---|---|---|---|---|---|---|
| **TOOL: Braden [8,9] (1987)** | | | | | | |
| Huang [41] (2021) | n = 60; N = 49,326 | **0.78 (0.74, 0.82)** [A] N = 15,241 By cut-off: *≤15* (n = 15): 0.79 (0.76, 0.82) *16* (n = 19): 0.75 (0.67, 0.82) *17* (n = 4): 0.69 (0.61, 0.76) *18* (n = 15): 0.82 (0.73, 0.89) *≥19* (n = 7): 0.78 (0.65, 0.87) | **0.72 (0.66, 0.78)** [A] N = 34,085 By cut-off: *≤15*: 0.66 (0.55, 0.75) *16*: 0.85 (0.70, 0.93) *17*: 0.86 (0.50, 0.97) *18*: 0.70 (0.62, 0.77) *≥19*: 0.54 (0.44, 0.63) | **PLR 2.80 (2.30, 3.50)** [A] **NLR 0.30 (0.26, 0.35)** [A] | **9.00 (7.00, 13.00)** [A] By cut-off: *≤15*: 7.00 (4.00, 12.00) *16*: 17.00 (8.00, 36.00) *17*: 14.00 (2.00, 103.00) *18*: 11.00 (6.00, 20.00) *≥19*: 4.00 (2.00, 7.00) | **0.82 (0.79, 0.85)** [A] By cut-off: *≤15*: 0.80 (0.76, 0.83) *16*: 0.84 (0.80, 0.87) *17*: 0.73 (0.69, 0.77) *18*: 0.83 (0.79, 0.86) *≥19*: 0.67 (0.63, 0.71) |
| Zhang [53] (2021) | n = 18; N = 11,167 | **0.78 (0.68, 0.85)** [B] | **0.61 (0.40, 0.79)** [B] | **PLR 2.00 (1.24, 3.24)** **NLR 0.36 (0.25, 0.52)** | 5.52 (2.61, 11.67) | 0.78 |
| Park [45] (2016b) | n = 25; N = 10,547 | **0.72 (0.69, 0.74)** [A] | **0.63 (0.62, 0.64)** [A] | **PLR 2.31 (1.98, 2.69)** [A] **NLR 0.43 (0.36, 0.51)** [A] | 6.50 (4.64, 9.11) [A] | **0.79** [A] (SE = 0.02) |
| **TOOL: Modified Braden scales: Braden–modified by Song and Choi [67] (1991)** | | | | | | |
| Park [44] (2016a) | n = 4; N = 688 | **0.97 (0.92, 0.99)** [A] N = 125 | **0.70 (0.66, 0.73)** [A] N = 563 | **PLR 3.47 (1.33, 9.06)** [A] **NLR 0.08 (0.04, 0.19)** [A] | 56.56 (21.88, 146.21) [A] | **0.95** [A] (SE 0.02) |
| **TOOL: Braden–modified by Pang and Wong [68] (1998)** | | | | | | |
| Park [44] (2016a) | n = 2; N = 626 | **0.89 (0.71, 0.98)** [A] N = 27 | **0.71 (0.67, 0.75)** [A] N = 599 | **PLR 2.87 (1.88, 4.38)** [A] **NLR 0.17 (0.06, 0.49)** [A] | 16.06 (4.75, 54.35) [A] | Not calculated |
| **TOOL: Cubbin and Jackson [69] (1991)** | | | | | | |
| Chen [36] (2023) | n = 9; N = 7,684 | **0.81 (0.51, 0.95)** N = 1,558 | **0.76 (0.58, 0.88)** N = 6,126 | **PLR 3.34 (2.14, 5.21)** **NLR 0.25 (0.09, 0.68)** | 13.24 (5.41, 32.40) | **0.84 (0.81, 0.87)** |
| Zhang [53] (2021) | n = 6; N = 800 | **0.84 (0.59, 0.95)** [B] | **0.84 (0.66, 0.93)** [B] | **PLR 5.12 (2.70, 9.70)** **NLR 0.19 (0.08, 0.49)** | 26.45 (13.51, 51.78) | 0.90 |
| Park [44] (2016a) | n = 4; N = 662 | **0.67 (0.60, 0.74)** [A] N = 194 | **0.75 (0.71, 0.79)** [A] N = 468 | **PLR 2.80 (1.66, 4.72)** [A] **NLR 0.34 (0.15, 0.76)** [A] | 9.46 (2.41, 37.22) [A] | **0.82** [A] (SE 0.06) |
| **TOOL: EVARUCI [70] (2001)** | | | | | | |
| Zhang [53] (2021) | n = 3; N = 3,063 | **0.84 (0.79, 0.89)** [A] | **0.68 (0.66, 0.70)** [A] | **PLR 2.32 (2.14, 2.51)** [A] **NLR 0.25 (0.19, 0.35)** [A] | 9.79 (6.81, 14.07) [A] | **0.82** [A] |
| **TOOL: Norton [10] (1962)** | | | | | | |
| Park [44] (2016a) | n = 7; N = 2,899 | **0.75 (0.70, 0.79)** [A] N = 383 | **0.57 (0.55, 0.59)** [A] N = 2,516 | **PLR 1.77 (1.26, 2.50)** [A] **NLR 0.49 (0.32, 0.76)** [A] | 7.57 (2.53, 22.64) [A] | **0.82** [A] (SE 0.05) |
| Park [45] (2016b) | n = 5; N = 2,408 | **0.76 (0.71, 0.80)** [A] | **0.55 (0.53, 0.57)** [A] | **PLR 1.58 (1.07, 2.34)** [A] **NLR 0.47 (0.29, 0.76)** [A] | 6.41 (1.72, 23.88) [A] | **0.84** [A] (SE 0.07) |
| **TOOL: Waterlow [11] (1985)** | | | | | | |
| Zhang [53] (2021) | n = 4; N = 1,000 | **0.63 (0.48, 0.76)** [B] | **0.46 (0.22, 0.71)** [B] | **PLR 1.16 (0.66, 2.01)** **NLR 0.82 (0.40, 1.67)** | 1.42 (0.40, 5.07) | 0.56 |
| Park [44] (2016a) | n = 6; N = 1,268 | **0.55 (0.49, 0.62)** [A] N = 246 | **0.82 (0.80, 0.85)** [A] N = 1,222 | **PLR 2.89 (1.74, 4.79)** [A] **NLR 0.46 (0.31, 0.70)** [A] | 9.22 (6.43, 13.23) [A] | **0.82** [A] (SE 0.03) |
| Park [45] (2016b) | n = 5; N = 1,406 | **0.53 (0.47, 0.60)** [A] | **0.84 (0.81, 0.86)** [A] | **PLR 3.09 (1.63, 5.83)** [A] **NLR 0.49 (0.34, 0.72)** [A] | 9.06 (6.30, 13.04) [A] | **0.81** [A] (SE 0.03) |
| **ML models:** [C] | | | | | | |
| Pei [47] (2023) **Various ML models** | n = 14; N = 328,789 | **0.79 (0.78, 0.80)** N = 18,807 | **0.87 (0.88, 0.87)** N = 309,982 | **PLR 10.71 (5.98, 19.19)** **NLR 0.21 (0.08, 0.50)** | 52.39 (24.83, 110.55) | 0.94 |
| Qu [48] (2022) **DT models** | n = 14; N = 118,292 | **0.66 (0.42, 0.84)** N = 7,557 | **0.90 (0.78, 0.96)** N = 110,735 | **PLR 6.9 (3.2, 14.7)** **NLR 0.37 (0.20, 0.69)** | 18 (7, 49) | **0.88 (0.85, 0.91)** |

*(Continued)*

**Table 3.** (Continued)

| Review author (publication year) | n studies; N participants | Sensitivity (95% CI) (N = no. participants with PI) | Specificity (95% CI) (N = no. participants without PI) | Likelihood ratios (95% CI) | DOR (95% CI) | AUROC (95% CI) |
|---|---|---|---|---|---|---|
| Qu [48] (2022) LR models | n = 14; N = 195,927 | 0.71 (0.60, 0.80) N = 9046 | 0.83 (0.75, 0.89) N = 186,881 | PLR 4.3 (3.1, 5.9) NLR 0.35 (0.26, 0.46) | 12 (9, 17) | 0.84 (0.81, 0.87) |
| Qu [48] (2022) NN models | n = 9; N = 97,815 | 0.73 (0.55, 0.86) N = 9488 | 0.78 (0.65, 0.87) N = 88,327 | PLR 3.3 (2.1, 5.0) NLR 0.35 (0.21, 0.59) | 9 (5, 19) | 0.82 (0.79, 0.85) |
| Qu [48] (2022) RF models | n = 7; N = 161,334 | 0.72 (0.26, 0.95) N = 5486 | 0.96 (0.80, 0.99) N = 155,848 | PLR 16.3 (2.4, 108.9) NLR 0.29 (0.07, 1.29) | 56 (3, 1258) | 0.95 (0.93, 0.97) |
| Qu [48] (2022) SVM models | n = 9; N = 152,068 | 0.81 (0.69, 0.90) N = 6562 | 0.81 (0.59, 0.93) N = 145,506 | PLR 4.3 (1.8, 9.9) NLR 0.23 (0.13, 0.39) | 19 (6, 54) | 0.88 (0.85, 0.90) |

[A] Summary statistic across multiple thresholds.

[B] Estimate derived from HSROC, but method for choosing summary point unclear.

[C] It is not reported, at review level, whether the results from the Qu review [48] include data from development, internal validation, or external validation/evaluation studies.

AUROC, area under the receiver operating characteristic curve; CI, confidence interval; DT, decision tree; DOR, diagnostic odds ratio; HSROC, hierarchical summary receiver operating characteristic curve; LR, logistic regression; ML, machine learning; NLR, negative likelihood ratio; NN, neural network; NS, not stated; PI, pressure injury; PLR, positive likelihood ratio; RF, random forest; SE, standard error; SVM, support vector machine.

**Norton scale.** Park and colleagues [44] synthesised data from 7 studies (2,899 participants) evaluating the Norton scale, across thresholds ranging from <14 to <16. They reported summary sensitivity of 0.75 (95% CI [0.70, 0.79]) and specificity 0.57 (95% CI [0.55, 0.59]). A further 4 reviews presented statistically synthesised results for the Norton scale (Appendix E in S1 Supporting Information), including one review by Chou and colleagues [38] which included 9 studies (5,444 participants) but only reported median values for accuracy parameters.

**Waterlow scale.** Although Zhang and colleagues [53] included the fewest participants (4 studies; 1,000 participants) of all 6 reviews that conducted a statistical synthesis of the accuracy of the Waterlow scale [11], they provided the most recent review. It was rated highest on AMSTAR-2 criteria and appropriately used the HSROC model for meta-analysis across thresholds ranging from 12 to 25. Summary sensitivity was 0.63 (95% CI [0.48, 0.76]) and summary specificity 0.46 (95% CI [0.22, 0.71]) (Table 3). A second review [44] reported contrasting results with summary sensitivity of 0.55 (95% CI [0.49, 0.62]) and specificity 0.82 (95% CI [0.80, 0.85]) (6 studies; 1,268 participants); however, authors synthesised data across multiple thresholds without utilising hierarchical methods.

**Machine learning algorithms.** Pei and colleagues [47] included 18 ML models, 7 of which were not covered by any other included review. Accuracy measures were combined across all models that provided 2 × 2 data (n = 14 models). The summary AUC across the 14 models was 0.94, summary sensitivity was 0.79 (95% CI [0.78, 0.80]), and summary specificity was 0.87 (95% CI [0.88, 0.87]) (Table 3). Meta-regression found no significant effect by ML algorithm or data type. Clinical heterogeneity was not investigated. The majority of studies (16/18, 89%) were considered at high risk of bias based on PROBAST. Our confidence in the review was critically low, with only 6/15 (40%) AMSTAR-2 criteria fulfilled. One critical flaw was the use of inappropriate meta-analysis methods (failing to use a hierarchical model for synthesising sensitivity and specificity).

Qu and colleagues [48] conducted separate meta-analyses of 25 studies by ML algorithm type using Bayesian hierarchical methods (Table 2). The review rated critically low on AMSTAR-2 items, with only 6/15 (40%) criteria fulfilled. The review did not restrict inclusion

to external evaluations of the models, and the authors did not report which estimates were sourced from development data or external data. The summary AUC for the 5 algorithms ranged from 0.82 (95% CI [0.79, 0.85]; 9 studies with 97,815 participants) for neural network-based models to 0.95 (95% CI [0.93, 0.97]; 7 studies with 161,334 participants) for random forest models (Table 3).

The latter approach also had the highest summary specificity 0.96 (95% CI [0.80, 0.99]), with sensitivity 0.72 (95% CI [0.26, 0.95]). The highest summary sensitivity was observed for support vector machine models (0.81, 95% CI [0.69, 0.90]) with summary specificity 0.81 (95% CI [0.59, 0.93]) (9 studies, 152,068 participants). The remaining algorithms had summary sensitivities ranging from 0.66 (decision tree models) to 0.73 (neural network models) (Table 3). Two additional ML algorithms evaluated in the included studies (Bayesian networks and LOS (abbreviation not explained)) had too few studies to allow meta-analysis (Appendix E in S1 Supporting Information).

**Other scales.** In addition to the risk prediction tools reported above, Zhang and colleagues [53] reported on the EVARUCI scale [70], presenting summary sensitivity and specificity of 0.84 (95% CI [0.79, 0.89]) and 0.68 (95% CI [0.66, 0.70]), respectively (3 studies; 3,063 participants). These results were synthesised across thresholds, 11 and 11.5 (one not reported).

Additional statistical syntheses covering 3 further modifications of the Braden scale (Braden modified by Kwong [71], the 4-factor model [72], and "extended Braden" [72]), 2 modified versions of the Norton scale (by Ek [73] and by Bienstein [74]), a revised "Jackson & Cubbin" [75], and the EMINA [76] and PSPS [77] tools) were also identified [38,39,49]. These analyses showed variable performance, often with high uncertainty. Full details can be found in Table D in S1 Supporting Information.

Table E in S1 Supporting Information reports data for another 17 risk prediction tools, each associated with a single primary study (therefore not covered in detail in the text above), and another 2 tools, Sunderland [78] and RAPS [79], which are assessed in 2 primary studies each.

## Results from reviews evaluating the clinical effectiveness of risk prediction tools

The 11 reviews reporting clinical effectiveness used a range of eligibility criteria and a number of different quality assessment tools, leading to varying conclusions about the methodological quality of the same studies across reviews. Given the overlap in study inclusion between reviews Table 4 provides an overview of results from 4 [38,57,59,61] of the 11 reviews, and a summary of the included comparative studies is provided below.

Two randomised controlled trials (RCTs) of risk prediction tools [83,84] were identified, both of which were considered at high risk of bias in the Cochrane review (assessed using the Cochrane RoB tool [64]). One of the trials (an individually randomised study [83]) was included in a further 3 reviews which considered it to be "good quality" [38], "valid" [56], or "high quality" [59]. The trial was conducted in 1,231 hospital inpatients and the only intervention was that the staff must use the tool that was allocated to them, with no other protocol prescribed changes made to routine care. However, no evidence of a difference in PI incidence was found between patients assessed with either the Waterlow scale or Ramstadius tool compared with clinical judgement alone (RR 1.10 (95% CI [0.68, 1.81]) and RR 0.79 (95% CI [0.46, 1.35]), respectively. The trial further showed no evidence of a difference in patient management or in PI severity when using a risk assessment tool compared to clinical judgement.

A further cluster randomised trial [84] was considered to be of poor methodological quality both in the Cochrane review [38] and one other review [61]. The trial included 521 patients at a military hospital and compared nurse training with mandatory use of the Braden scale, to

**Table 4. Systematic reviews evaluating clinical effectiveness.**

| Review author (publication year) | Tools included | Setting of included studies; study design; sample size | Included outcomes | Brief description of study quality | Relevant results from included studies |
|---|---|---|---|---|---|
| Lovegrove [59] (2021) | Braden; Maelor score; Norton; Ramstadius; Waterlow | Acute care hospital n = 1, inpatient units n = 1, ICU n = 1, internal medicine and oncology wards n = 1 Design: cross-sectional survey n = 2, RCT n = 1, observational inter-rater reliability n = 1; Sample size 45 to 1,231. | PI risk scores; PI incidence; PI preventive interventions; inter-rater reliability (reliability results covered in Appendix E in S1 Supporting Information) | RoB assessed using JBI tools or analytical cross-sectional study appraisal checklist. The RCT was judged as high quality. Of the remaining studies, 2 were judged as high quality and one as moderate quality; inclusion criteria not clearly stated and no strategies to deal with confounding. | • There were no differences in patient management ("pressure care plan" and use of a special mattress) based on PI risk assessment method (clinical judgement, Ramstadius tool, or Waterlow score). PI incidence difference between groups not significant (p = 0.44) (Webster 2011 [83]). • A hospital that used the Maelor scale reported a higher rate of PI preventive strategies, and a lower PI prevalence (12% vs. 54%) than a site that used nurses' clinical judgement (Moore 2015 [85]). |
| Moore [61] (2019) | Braden; Waterlow; Ramstadius | Military hospital n = 1, internal medicine and oncology wards n = 1; Design: RCT n = 1, cluster randomised trial n = 1; Sample sizes 286 and 1,231. | PI incidence; severity of PIs | RoB assessed using Cochrane tool (Higgins 2011 [64]). Both studies at high RoB due to blinding issues. One study at RoB also due to baseline imbalance and incorrect analyses. | • No differences in PI incidence when using Braden scale or clinical judgement (Braden vs. clinical judgement +training, RR 0.97 (95% CI [0.53, 1.77]); Braden vs. clinical judgement RR 1.43 (95% CI [0.77, 2.68]) (Saleh 2009[84]). • No difference in PI incidence when using a risk assessment tool compared to clinical judgement (RR 1.10 (95% CI [0.68, 1.81]) and RR 0.79 (95% CI [0.46, 1.35]), for Waterlow and Ramstadius, respectively) (Webster 2011 [83]). • No difference in PI severity based on risk assessment tools vs. clinical judgement (Webster 2011[83]). |
| Chou [38] (2013) | Norton modified by Bale; Braden; Waterlow; Ramstadius | Hospital n = 2, hospice n = 1; Design: non-randomised n = 1, cluster randomised trial n = 1, RCT n = 1; Sample size 240 to 1,231. | PI incidence, severity of PIs; PI preventive interventions | RoB assessed with criteria consistent with AHRQ Methods Guide for Effectiveness and Comparative Effectiveness Reviews. One RCT was rated as good quality and the other as poor due to randomisation and blinding issues. The cohort study was rated as poor; there were blinding issues and confounding was not investigated. | • No difference in PI incidence when using a risk assessment tool compared to clinical judgement (RR 1.10 (95% CI [0.68, 1.81]) and RR 0.79 (95% CI [0.46, 1.35]), for Waterlow and Ramstadius, respectively) (Webster 2011 [83]).• The modified version of the Norton scale with use of preventive interventions is associated with lower risk of PIs compared with clinical judgement (RR 0.11, 95% CI [0.03, 0.46]) (Bale 1995 [88]). • No difference in risk of PIs when one of 3 interventions was used (22% vs. 22% vs. 15%, p = 0.38 for nurse training +mandatory Braden scale, nurse training+optional Braden scale and no training, respectively) (Saleh 2009 [84]). |

(*Continued*)

**Table 4.** (Continued)

| Review author (publication year) | Tools included | Setting of included studies; study design; sample size | Included outcomes | Brief description of study quality | Relevant results from included studies |
|---|---|---|---|---|---|
| Health Quality Ontario [57] (2009) | Norton; Norton modified by Bale; Norton modified by Ek 97 | Hip fracture inpatients $n = 1$, palliative care/hospice $n = 1$, neurosurgery, general medicine, orthopaedic, and oncology units $n = 1$; Design: prospective controlled (contemporaneous controls) $n = 1$, before-and-after $n = 1$; Sample size 124 to 223. | PI incidence; PI preventive interventions | RoB assessment criteria name not given. Two studies met 6/8 and one study met all quality assessment requirements. In the studies that did not meet all requirements, there were blinding and loss to follow-up issues. One study used a version of the Norton scale that was not validated. | • Compared a strategy that gave high-risk patients (based on modified Norton score) a risk alarm sticker to standard care. No significant difference between the groups in the incidence of PIs (Gunningberg 1999 [87]). • Compared a strategy where patients received a pressure support system allocated according to the modified Norton scale to one where the nurse chose whether to give a special mattress. Using the scale significantly reduced the incidence of PIs (22.4% vs. 2.5%, $p < 0.001$) (Bale 1995 [88]). • Compared the Norton scale with training to standard care. There was a significant difference in the number of preventive interventions (18.96 vs. 10.75, for Norton and usual care, respectively). Interventions were used earlier for Norton vs. usual care (on day 1, 61% vs. 50%, $p < 0.002$). No significant difference in the incidence of PIs between the groups (Hodge 1990 [89]). |

AHRQ, Agency for Healthcare Research and Quality; CI, confidence interval; ICU, intensive care unit; JBI, Joanna Briggs Institute; PI, pressure injury; RCT, randomised controlled trial; RoB, risk of bias; RR, risk ratio.

nurse training and optional use of the Braden scale, to no training. No evidence of a difference in PI incidence was observed between the 3 groups: incidence rates were 22%, 22%, and 15% ($p = 0.38$), respectively.

Two reviews by Lovegrove and colleagues [59,60] included an uncontrolled comparison study [85] rated as high quality [59]. The study compared the clinical effectiveness of the Maelor scale [86] used in an Irish hospital (121 patients) with nurses' clinical judgement at a Norwegian hospital (59 patients). A higher rate of preventive strategies, as well as a lower PI prevalence (12% versus 54%), was reported for the Irish hospital. However, these results are likely to be highly confounded by inherent differences in population and setting.

A non-randomised study by Gunningberg and colleagues [87] included in 2 reviews [43,57] was considered by review authors to be of relatively high quality. The study was conducted in 124 patients in emergency and orthopaedic units and compared the use of a PI risk alarm sticker for patients with a modified Norton Score of <21 (indicating high-risk patients) to standard care. No significant difference in the incidence of PIs between the Norton scale and standard care groups was observed.

A non-randomised study [88] conducted in 233 hospice inpatients was included in three reviews [38,43,57], one of which is reported in Table 4 [57]. The study met 6 of 8 quality criteria

used by Health Quality Ontario [57]. Use of a modified version of the Norton scale (Norton modified by Bale), in conjunction with standardised use of preventive interventions based on risk score, was found to be associated with lower risk of PIs when compared with nurses' clinical judgement alone (RR 0.11, 95% CI [0.03, 0.46]). The lack of randomisation limits the reliability of this result, and review authors report that the modified Norton scale had not been validated.

Finally, a "before-and-after" study [89] of 181 patients in various hospital settings was included in 2 reviews [43,57], one of which considered the study to meet all quality criteria [57]. Use of the Norton scale with additional training for staff was associated with significant differences in the number of preventive interventions prescribed compared to standard care (18.96 versus 10.75, respectively). Preventive interventions were also introduced earlier in the intervention group (on day 1, 61% versus 50%, $p < 0.002$ for Norton and usual care, respectively). However, no significant difference in the incidence of PIs was detected between the groups.

## Discussion

This umbrella review summarises data from a total of 26 systematic reviews of studies evaluating the prognostic accuracy and clinical effectiveness of a total of 70 PI risk prediction tools. Despite the large number of available reviews, quality assessment using an adaptation of AMSTAR-2 suggested that the majority were conducted to a relatively poor standard or did not meet reporting standards for systematic reviews [19,90]. Of the 15 AMSTAR-2 items assessed, only 2 (for accuracy reviews) and 4 (for effectiveness reviews) criteria were more consistently met (more than 60% of reviews scoring "Yes"). While AMSTAR-2 Item 6 (data extraction independent in duplicate) was fulfilled by over half of all reviews (15/26, 58%), and Item 14 (adequate heterogeneity investigation) was fulfilled by around half of the accuracy reviews (10/19, 53%), all other criteria were fully met by less than half of the reviews. The primary studies included in the reviews were particularly poorly described in the accuracy reviews, making it difficult to determine exactly what was evaluated and in whom. The extent to which we could reliably describe and comment on the content of the reviews is limited and high-quality evidence for the accuracy and clinical effectiveness of PI risk prediction tools may be lacking.

Of the 19 reviews reporting the accuracy of included tools, only 2 used appropriate methods for both quality assessment and statistical synthesis of accuracy data [41,53], one of which [41] evaluated only the Braden scale. Only 2 reviews [42,43] prespecified the exclusion of studies reporting accuracy data from tool development studies, one review restricted to "validated risk assessment instruments" only [38] and one review [47] was limited to development studies only. This was the only review [47] that discussed the importance of appropriate validation of prediction tools. Only 2 reviews conducted meta-analyses at different cut-offs for determination of high risk [38,41]; the remaining reviews combined data regardless of the threshold used. Combining data across different thresholds to estimate summary sensitivity and specificity yields clinically uninterpretable and non-generalisable estimates that do not relate to a particular threshold [35]. Only 1 review [38] considered timing in their inclusion criteria or in the description of primary studies. It is important to interpret the findings below with these limitations in mind.

The included meta-analyses suggested that risk prediction scales have moderate sensitivities and somewhat lower specificities, typically in the range of around 60% to 85% for sensitivity and 50% to 80% for specificity. Although these ranges in sensitivities and specificities would be considered on the lower end of acceptable within a diagnostic test accuracy (DTA) paradigm, they may have greater utility in a prognostic context. Without a detailed review of the primary study publications for these tools, it is not possible to assess which, if any, of these risk

assessment scales might outperform the others. It seems that limited comparative studies comparing the accuracy of different tools are available.

For the ML-based models, one review [47] combined multiple ML models into 1 meta-analysis and another [48] meta-analysed accuracy data by algorithm type. The results of the latter meta-analyses are not informative for clinical practice but may be a useful way of identifying which ML algorithms may be more suited to the data. Results suggested that specificities for random forest or decision tree models could reach 90% or above with associated sensitivities in the range of 66% to 72%; however, relatively wide confidence intervals around these summary estimates reflect considerable variation in model performance. Moreover, some of these estimates came from internal validations within model development studies and may not be transferable to other settings [91]. Authors should make it clear where accuracy estimates are derived from to avoid overinterpretation of results.

Diagnostic accuracy studies are typically cross-sectional in the sense that there should be no, or only minimal delay between application of the test and the reference standard [92,93]. For prognostic accuracy however, there is a time delay between the application of the test and the outcome that the tool aims to predict. If the use of an accurate PI risk prediction tool is combined with effective and appropriate preventive measures in those identified as most at risk, the incidence of PI would decline, reducing the positive predictive value of the original risk assessment and potentially the sensitivity of the tool [94]. Sensitivity and specificity can be optimised by methods which directly consider the cost of misclassification, including both the harms associated with applying more intensive prevention in those with a false positive result and the benefits of preventive measures in those with a true positive result. One solution to determine the preventive treatment threshold risk is through net benefit calculations [95,96], which can be visualised in decision curves and are common in prognostic research. These calculations can assist in providing a balanced use of resources while maximising positive health outcomes, such as lowering incidence of PI.

It is important to also consider that not all predictors have a causal relationship with the outcome, therefore, not every predictor will be a clinical risk modifier. Risk assessment tools that allow a more personalised-risk approach, i.e., that identify and flag predictors that are risk modifiers to the end-users of the tool would make predictions more interpretable and actionable. Some such developments exist [97,98], but future validation of these methods is needed. Where risk assessment tools are developed for enriching study design (for example, as a means of recruiting only high-risk patients to studies evaluating preventive measures), a different approach and optimisation of performance metrics would be needed. Risk prediction models should therefore prespecify their intended application before development to allow their clinical utility for a given context to be addressed [99].

Prediction models, like any test used for diagnostic or prognostic purposes, require evaluation in the care pathway to identify the extent to which their use can impact on health outcomes [100]. Of the 11 reviews assessing clinical effectiveness of PI risk prediction tools, the only primary studies suggesting potential patient benefits from the use of risk prediction tools [85,88,89] were non-randomised and are likely to be at high risk of bias. In contrast, 2 randomised trials [83,84] (both considered at high risk of bias by the Cochrane review [61]) suggest that use of structured risk assessment tools does not ultimately lead to the reduction in incidence of PIs. We should recognise that effectiveness outcomes from using a risk prediction tool depend on the timely implementation of effective preventive measures, a step that is frequently poorly described in studies evaluating the effectiveness of risk assessment tools, restricting the conclusions that can be drawn from the limited evidence available. One possible explanation for the lack of differences in PI incidence is the implementation of preventive measures that have not been proven effective in preventing PIs, such as alternating air

mattresses [4]. All reviews included studies that assessed the use of risk assessment scales developed by clinical experts, and no evidence is available evaluating the clinical effectiveness of empirically derived prediction models or ML algorithms.

We have separately reviewed [7] available evidence for the development and validation of risk prediction tools for PI occurrence. Almost half (60/124, 48%) of available tools were developed using ML methods (as defined by review authors), 37% (46/124) were based on clinical expertise or unclear methods, and only 15% (18/124) were identified as having used statistical modelling methods. The reviews varied in methodological quality and reporting; however, the reporting of prediction model development in the original primary studies appears to be poor. For example, across all prediction tools identified, the internal validation approach was unclear and unidentifiable for 72% (89/124) of tools, and only 2 reviews [47,101] identified and included external validation studies ($n$ = 9 studies).

ML-based models may have potential for identifying those at risk of PI, as suggested by 2 reviews [47,48] included in this umbrella review. However, it is important to consider the lack of transparency in reporting of model development methods and model performance, and the concerning lack of model validation in populations outside of the original model development sample [7].

We have conducted the first umbrella review that summarises the prognostic accuracy and clinical effectiveness of prediction tools for risk of PI. We followed Cochrane guidance [18], with a highly sensitive search strategy designed by an experienced information specialist. Although we excluded non-English publications due to time and resource constraints, where possible these publications were used to identify additional eligible risk prediction tools.

To some extent, our review is limited by the use of AMSTAR-2 for quality assessment of included reviews. AMSTAR-2 was not designed for assessing systematic reviews of diagnostic or prognostic studies. Although we made some adaptations, many of the existing and amended criteria relate to the quality of reporting of the reviews as opposed to methodological quality. There is scope for further work to establish criteria for assessing systematic reviews of prediction tools. Additionally, we chose not to exclude reviews based on low AMSTAR-2 ratings to provide a comprehensive overview of all available evidence. However, by doing so, we acknowledge that many included reviews are of poor quality (with critically low confidence in 81%, 21/26, reviews), reducing the reliability of the evidence presented and the ability to make conclusions or recommendations based on this evidence.

The primary limitation of our study lies in the limited detail available on risk prediction tools and their performance within the included systematic reviews. To ensure comprehensive model identification, we adopted a broad definition of "systematic," potentially influencing the depth of information provided in the reviews, and the reporting quality in many primary studies contributing to these reviews may be suboptimal.

Although standards for reporting of test accuracy studies have been available since the year 2000 [92], standards for reporting risk prediction models were not published until 2015 [102]. Similarly, quality assessment tools highlighting important areas for consideration in primary studies have been available for DTA studies since 2003, with an adaption to prognostic accuracy published in 2022 [103] and PROBAST for prediction model studies in 2019 [33]. This lag in methodological developments for studies and systematic reviews of risk prediction tools has likely contributed to the observed emphasis on the application of DTA principles in our set of reviews, without sufficient consideration of the prognostic context and effect on accuracy of intervening and effective preventive interventions.

While 95% (18/19) accuracy reviews aimed to evaluate the "predictive" validity of PI risk assessment tools, the majority (16/19, 84%) relied on DTA principles without any consideration of the time interval between the test and the outcome, i.e., occurrence of PI. This

approach does not account for the prognostic nature of these tools or address longitudinal questions, such as censoring and competing events [103]. Another fundamental flaw in these accuracy assessments is that risk scales may actually appear to perform worse in settings where risk prediction and preventive care are most effective, as accurate risk prediction combined with effective preventive measures may prevent patients classified as "high-risk" from developing PIs [94].

In conclusion, this umbrella review comprehensively summarises the prognostic accuracy and clinical effectiveness of risk prediction tools for developing PIs. The included systematic reviews used poor methodology and reporting, limiting our ability to reliably describe and evaluate their content. ML-based models demonstrated potential, with high specificity reported for some models. Wide confidence intervals highlight the variability in current evaluations, and external validation of ML tools may be lacking. The prognostic accuracy of clinical scales and statistically derived prediction models has a substantial range of specificities and sensitivities, motivating further model development with high-quality data and appropriate statistical methods.

Regarding clinical effectiveness, a reduction of PI incidence is unclear due the overall uncertainty and potential biases in available studies. This underscores the need for further research in this critical area, once promising prediction tools have been developed and appropriately validated. In particular, the clinical impact of newer ML-based models currently remains largely unexplored. Despite these limitations, our umbrella review provides valuable insights into the current state of PI risk prediction tools, emphasising the need for robust research methods to be used in future evaluations.

## Supporting information

**S1 Supporting Information. Appendices.**
(PDF)

## Acknowledgments

We would like to thank Mrs. Rosie Boodell (University of Birmingham, UK) for her help in acquiring the publications necessary to complete this piece of work.

Paul Hartmann AG is part of HARTMANN GROUP. The contract with the University of Birmingham was agreed on the legal understanding that the authors had the freedom to publish results regardless of the findings.

The views expressed are those of the authors and not necessarily those of the NIHR or the Department of Health and Social Care.

## Author Contributions

**Conceptualization:** Bethany Hillier, Katie Scandrett, April Coombe, Tina Hernandez-Boussard, Ewout Steyerberg, Yemisi Takwoingi, Vladica M. Veličković, Jacqueline Dinnes.

**Data curation:** Bethany Hillier, Katie Scandrett, April Coombe, Jacqueline Dinnes.

**Formal analysis:** Bethany Hillier, Katie Scandrett, Jacqueline Dinnes.

**Funding acquisition:** Yemisi Takwoingi, Vladica M. Veličković, Jacqueline Dinnes.

**Investigation:** Bethany Hillier, Katie Scandrett, April Coombe, Yemisi Takwoingi, Jacqueline Dinnes.

**Methodology:** Bethany Hillier, Katie Scandrett, April Coombe, Tina Hernandez-Boussard, Ewout Steyerberg, Yemisi Takwoingi, Vladica M. Veličković, Jacqueline Dinnes.

**Project administration:** Bethany Hillier, Yemisi Takwoingi, Jacqueline Dinnes.

**Resources:** Bethany Hillier, Katie Scandrett, Yemisi Takwoingi.

**Supervision:** Yemisi Takwoingi, Jacqueline Dinnes.

**Writing – original draft:** Bethany Hillier, Katie Scandrett, April Coombe, Jacqueline Dinnes.

**Writing – review & editing:** Bethany Hillier, Katie Scandrett, April Coombe, Tina Hernandez-Boussard, Ewout Steyerberg, Yemisi Takwoingi, Vladica M. Veličković, Jacqueline Dinnes.

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
