## [Editor Report · Decision Letter 0]

12 Mar 2024

Dear Dr Dinnes, 

Thank you for submitting your manuscript entitled "Accuracy and clinical effectiveness of risk prediction tools for pressure injury occurrence: An umbrella review" for consideration by PLOS Medicine.

Your manuscript has now been evaluated by the PLOS Medicine editorial staff and I am writing to let you know that we would like to send your submission out for external peer review. We are aware that the intended companion manuscript (PMEDICINE-D-24-00796) was returned to you in error. We were still able to access and read it and feel that it would be preferable to submit one complete paper.

Please re-submit your manuscript within two working days, i.e. by Mar 14 2024.

Feel free to email me at aschaefer@plos.org or us at plosmedicine@plos.org if you have any queries relating to your submission.

Kind regards,

Alexandra Schaefer, PhD

Associate Editor

PLOS Medicine

---

## [Decision Letter · Decision Letter 1]

12 Jun 2024

Dear Dr. Dinnes,

Many thanks for submitting your manuscript "Accuracy and clinical effectiveness of risk prediction tools for pressure injury occurrence: An umbrella review” (PMEDICINE-D-24-00795R1) to PLOS Medicine. The paper has been reviewed by four subject experts and a statistician; their comments are included below and can also be accessed here: 

[LINK]

As you will see, the reviewers were positive about the paper but, they raised a number of questions about specific study details and the methodological approach. After discussing the paper with the editorial team I’m pleased to invite you to revise the paper in response to the reviewers’ comments. We plan to send the revised paper to some of all of the original reviewers*, and of course we cannot provide any guarantees at this stage regarding publication.

When you upload your revision, please include a point-by-point response that addresses all of the reviewer and editorial points, indicating the changes made in the manuscript and either an excerpt of the revised text or the location (eg: page and line number) where each change can be found. Please submit a clean version of the paper as the main article file and a version with changes marked should as a marked-up manuscript. Please also check the guidelines for revised papers at http://journals.plos.org/plosmedicine/s/revising-your-manuscript for any that apply to your paper.

We ask that you submit your revision by July 3rd 2024. However, if this deadline is not feasible, please contact me by email, and we can discuss a suitable alternative.

Please don’t hesitate to contact me directly with any questions (pdodd@plos.org). If you reply directly to this message, please be sure to ‘Reply All’ so your message comes directly to my inbox.

Kind regards,

Pippa

Philippa Dodd Senior Editor, PLOS Medicine - on behalf of Alexandra Tosun, PhD

PLOS Medicine

plosmedicine.org

pdodd@plos.org

*Please note: If your article is accepted, you may have the opportunity to make the peer review history publicly available. The record will include editor decision letters (with reviews) and your responses to reviewer comments. If eligible, we will contact you to opt in or out.

Editorial comments:

1) We really enjoyed reading your paper and were pleased to see that it was also well received by the reviewers. We agree that it is not without limitations as currently written and require that you address each of the editorial and reviewer points detailed below in full. 

Specifically, we agree with the statistical reviewer that a statistical analysis of the evidence would be beneficial. Ideally “researchers should use the study specific data extracted from each SRMA to repeat each meta-analysis separately rather than report the meta-analytical result as presented in the original SRMA” ( see, https://bmjmedicine.bmj.com/content/1/1/e000071 for reference).

2) Many of the editorial requests detailed below pertain to specific formatting and content requirements. Some may have already been incorporated into the manuscript and some may not apply but please review the complete list of items and ensure that each item is included as necessary.

3) Reporting guidance 

Thank you for reporting your study in line with PRISMA-DTA guidance and for including the checklist. Please amend the checklist to refer to section and paragraph numbers as opposed to line numbers as these often change at publication. 

Please ensure that your search is updated to the current time. PLOS Medicine requires that searches are updated to within roughly 6 months of a potential publication date.

4) Statistical reporting 

Please quantify the main results with 95% CIs and p values. 

When reporting p values please report as <0.001 and where higher as p=0.002, for example. When reporting 95% CIs please separate upper and lower bounds with commas instead of hyphens as the latter can be confused with reporting of negative values. 

Please include the actual amounts and/or absolute risk(s) of relevant outcomes (including NNT or NNH where appropriate), not just relative risks or correlation coefficients. (example for absolute risks: PMID: 28399126). 

Please include any important dependent variables that are adjusted for in the analyses. 

5) Abstract layout 

Please structure your abstract using the PLOS Medicine headings (Background, Methods and Findings, Conclusions). 

6) Author summary 

At this stage, we ask that you include a short, non-technical Author Summary of your research to make findings accessible to a wide audience that includes both scientists and non-scientists. The authors summary should consist of 2-3 succinct bullet points under each of the following headings: 

• Why Was This Study Done? Authors should reflect on what was known about the topic before the research was published and why the research was needed. 

• What Did the Researchers Do and Find? Authors should briefly describe the study design that was used and the study’s major findings. Do include the headline numbers from the study, such as the sample size and key findings. 

• What Do These Findings Mean? Authors should reflect on the new knowledge generated by the research and the implications for practice, research, policy, or public health. Authors should also consider how the interpretation of the study’s findings may be affected by the study limitations. In the final bullet point of ‘What Do These Findings Mean?’, please describe the main limitations of the study in non-technical language. 

The Author Summary should immediately follow the Abstract in your revised manuscript. This text is subject to editorial change and should be distinct from the scientific abstract. Please see our author guidelines for more information: https://journals.plos.org/plosmedicine/s/revising-your-manuscript#loc-author-summary

7) Introduction layout 

Please address past research and explain the need for and potential importance of your study. Indicate whether your study is novel and how you determined that. If there has been a systematic review of the evidence related to your study (or you have conducted one), please refer to and reference that review and indicate whether it supports the need for your study. 

8) Discussion layout 

Please present and organize the Discussion as follows: a short, clear summary of the article's findings; what the study adds to existing research and where and why the results may differ from previous research; strengths and limitations of the study; implications and next steps for research, clinical practice, and/or public policy; one-paragraph conclusion. 

9) Supporting information 

Please cite your Supporting Information as outlined here: https://journals.plos.org/plosmedicine/s/supporting-information

In the published article, supporting information files are accessed only through a hyperlink attached to the captions. For this reason, you must list captions at the end of your manuscript file. You may include a caption within the supporting information file itself, as long as that caption is also provided in the manuscript file. Do not submit a separate caption file.

Comments from the reviewers:

Reviewer #1: Good to go

Reviewer #2: Thank you for the opportunity to review this manuscript. It highlights the ongoing problems/ lack of clarity with studies relating to predictive tools in PIs. 

Some of my comments are below. They are very minor and primarily relating to clarifications of stated numbers in the manuscript.

Line 111 - It is a bit unclear which author conducted the screening, i.e., which author plays the third reviewer role.

Line 156 - It is stated a total of 110, but Fig.1 indicates 111?

Line 189 - Just confirming if the mention of acute care here also includes hospital. If not, would the effectiveness review which stated 4/10, actually be 2? If yes, then the accuracy review would be 7/16, not 5/16.

Line 221 - Just checking the reported figure, i.e., should conflict of interest be 18/23 (not 17/23)?

Figure 2 has been crafted very nicely, and is very easy to read. Well done.

Reviewer #3: First of all, I would like to express my appreciation to the authors for conducting this meaningful study. Pressure injuries are a critical aspect of clinical care, and there is a huge number of studies on this topic. Thus, synthesizing existing information is highly valuable. This study provides a wealth of information on pressure injuries and offers valuable insights and essential information to readers. While the topic is very interesting and relevant, there are some areas that need improvement.

This paper provides a large amount of information, which seems to diminish the emphasis on the primary objective of the research. Given the nature of an umbrella review, it is appropriate to assess the quality of the papers reviewed. However, this aspect occupies a large amount of the content (e.g., Brief description of included study quality in Table 2). I suggest summarizing these evaluations or highlight the main objective of the study more effectively.

1. The terminology for pressure injury is used inconsistently throughout the paper, with terms like "pressure injury", "pressure ulcers,", and "PrI". Please standardize the terminology for clarity.

2. Table 1 contains several ambiguous expressions that need clarification.

- [Presents of pressure injury at baseline]: What is the definition of "baseline"? Does this refer to the presence of pressure injuries at the start of the study?

- [Setting]: Aren't "hospital" and "acute care (including surgical and ICU)" overlapping categories?

- In the Eligibility section, please replace the term "any" with a more precise/clear expression.

- [Source of data]: There seems to be an overlap between "Prospective only" and "prospective OR retrospective" as both include "prospective." Please verify whether it should be "prospective AND retrospective."

3. According to the Appendix, only Google Scholar limited the search period to 2003-2013. I would recommend to mention this period in the methodology section of the manuscript.

4. On page 18, it is mentioned that there are additional reviews ("The

341 individually randomized trial was included in three additional reviews29 44 47 50, each of which 342 considered the trial to be 'good quality'29, 'valid'44, or 'high quality'50). However, four references (29 44 47 50) are provided. It seems that reference 47 might be inappropriately included in the manuscript, and it should be verified.

5. While I do not recommend adding more content, I suggest considering the inclusion of this information in the author's decision. It may be beneficial to provide definitions or information of the representative pressure ulcer assessment tools (Braden, Norton, and Waterlow). Offering readers an understanding of the purpose and differences of these tools would enhance the practical application of the study, enabling them to identify the most effective and valid tool for their needs.

6. The paper focuses on the technical aspects of the major findings, while lacking sufficient clinically based explanations. Would there be a reason for this approach? For instance, if the Braden scale does not reduce the incidence of falls, there is a need for clinical interpretation of this result or a discussion on what aspects need improvement. If providing such an analysis is challenging, it could be included as a limitation of the study.

Reviewer #4: This paper evaluates the prognostic accuracy and clinical effectiveness of risk prediction tools for pressure injuries (PIs). Through an umbrella review of 16 systematic reviews on prognostic accuracy and 10 on clinical effectiveness, it was found that commonly used tools like the Braden, Norton, and Waterlow scales have variable sensitivities (53%-97%) and specificities (46%-84%). Only two randomized trials examined the clinical effectiveness of the tools, and showed no significant impact on PI incidence. However, both of the trials were assessed as having high risk of bias. The study highlights the need for high-quality research to develop and validate more effective PI risk prediction tools. Below are my specific comments on the paper. 

1. I am concerned about the potential conflict of interest in this paper. The work was commissioned by Paul Hartmann AG (PHAG), a company specializing in surgical and medical instruments, and three of the eight authors have close ties with PHAG. It is noted that VV is an employee of PHAG, but this is not reflected in the affiliations listed in the paper. I believe the conflict-of-interest issues in this paper require careful scrutiny.

2. Introduction, line 76-82: Comment: the introduction briefly mentions the limitations of current risk prediction tools but could delve deeper into these issues to underscore the necessity for this study. For instance, the authors could elaborate on specific methodological flaws in existing models and why these flaws justify the need for the current review study.

3. Method, lines 105-108: The authors searched MEDLINE, Embase, and CINAHL Plus EBSCO. However, I am curious why the Cochrane Library and Web of Science were not included in the literature search. Both are important databases for medical literature and should be considered.

4. Method, lines 105-108: Additionally, I believe that a search conducted in January 2023 may be somewhat outdated. I suggest the authors conduct an updated search to include the most recent literature.

5. Method, Lines 134-137: Could the authors elaborate on how they adapted AMSTAR-2 for the systematic review of risk prediction models in the main text? I believe a brief description is needed here. Additionally, is there any reference for the adaptation the authors made?

6. Method, Lines 138-153: The authors used a narrative synthesis approach with priority given to systematic reviews meeting certain criteria. I am curious why the authors did not conduct any statistical synthesis for tools that are examined across multiple studies. I believe that some statistical synthesis of the evidence for commonly used tools could be quite helpful.

Reviewer #5: Thank you for allowing me to review your excellent manuscript. Please see attached for my comments and recommendations.

[LINK]

---

## [Decision Letter · Decision Letter 2]

6 Nov 2024

Dear Dr. Dinnes,

Thank you very much for re-submitting your manuscript "Accuracy and clinical effectiveness of risk prediction tools for pressure injury occurrence: An umbrella review" (PMEDICINE-D-24-00795R2) for review by PLOS Medicine.

Thank you for your detailed response to the editors' and reviewers' comments. I have discussed the paper with my colleagues, and it has also been seen again by two of the original reviewers. The changes made to the paper were mostly satisfactory to the reviewers. As such, we intend to accept the paper for publication, pending your attention to the reviewers' and editors' comments below in a further revision. When submitting your revised paper, please once again include a detailed point-by-point response to the editorial comments.

[LINK]

In revising the manuscript for further consideration here, please ensure you address the specific points made by each reviewer and the editors. In your rebuttal letter you should indicate your response to the reviewers' and editors' comments and the changes you have made in the manuscript. Please submit a clean version of the paper as the main article file. A version with changes marked must also be uploaded as a marked up manuscript file. Please also check the guidelines for revised papers at http://journals.plos.org/plosmedicine/s/revising-your-manuscript for any that apply to your paper. 

We ask that you submit your revision within 1 week (Nov 13 2024). However, if this deadline is not feasible, please contact me by email, and we can discuss a suitable alternative.

Please do not hesitate to contact me directly with any questions (atosun@plos.org). If you reply directly to this message, please be sure to 'Reply All' so your message comes directly to my inbox.

We look forward to receiving the revised manuscript.

Sincerely,

Alexandra Tosun, PhD

Associate Editor 

PLOS Medicine

plosmedicine.org

Comments from Reviewers:

Reviewer #4: I believe the authors have sufficiently addressed and responded to both my and the other reviewers' comments and questions. I am satisfied with the revisions made to the manuscript and find the responses to be thorough. As a result, I consider the paper to be acceptable for publication.

Reviewer #5: (see also comments in the attachment)

This is an excellent paper and should generate discussion within the international pressure injury/ulcer community. To that end, I wrote a commentary both applauding your work but also identifying unanswered questions. Are we asking the right questions? I don't expect a response with solutions. You have beautifully summarized "what we know". I am asking "what do we still need to know?" Please see attached for Reviewer Commentary.

I recommended acceptance for publication with minor revisions. My source of continued concern can be found in lines 115-133 of the manuscript. Lines 115-129 provide a historic perspective of the initial development of the major risk assessment tools. This is fine....but it is limited in scope and sample size and fails to mention that there is a substantial body of literature evaluating the psychometric properties of these tools. Line 130 draws an immediate negative inference about the tools in stating "Despite the apparent lack of reporting of now standard methods for development....." I had difficulty finding Reference 7 in PubMed or Medline but eventually found it in EMBASE. I noted that preprint I located through EMBASE ....was not certified for peer review.

I have 3 concerns:

1. The negative inference mentioned above...

2. The reference to "now standard methods" which are not explained and were not available when psychometric testing of these tools was conducted.

3. The emphasis on lack of external validity within the context of these tools. This may apply to ML-generated risk prediction tools, but it doesn't really apply to tools that have been tested in thousands of clinical settings world-wide.

My recommendations is to reorder/revise the text starting at line 130 as follows....

There is a considerable body of evidence evaluating the psychometric properties and clinical utility of available risk prediction tools, much of which has been synthesised in systematic reviews and meta-analyses (7). However, there is an apparent lack of reporting of now standard methods for development and validation of risk prediction tools.

[LINK]

Requests from Editors:

GENERAL

The manuscript contains many abbreviations, please check carefully throughout the main text to ensure that abbreviations have been introduced appropriately.

CODE AVAILABILITY

We expect all researchers with submissions to PLOS in which author-generated code underpins the findings in the manuscript to make all author-generated code available without restrictions upon publication of the work. In cases where code is central to the manuscript, we may require the code to be made available as a condition of publication. Authors are responsible for ensuring that the code is reusable and well documented. Please make any custom code available, either as part of your data deposition or as a supplementary file. Please add a sentence to your data availability statement regarding any code used in the study, e.g. "The code used in the analysis is available from Github [URL] and archived in Zenodo [DOI link]" Please review our guidelines at https://journals.plos.org/plosmedicine/s/materials-software-and-code-sharing and ensure that your code is shared in a way that follows best practice and facilitates reproducibility and reuse. Because Github depositions can be readily changed or deleted, we encourage you to make a permanent DOI'd copy (e.g. in Zenodo) and provide the URL.

ABSTRACT

1) Please provide the beginning and end dates of your search.

2) Please provide the eligibility criteria and types of study designs included.

3) We suggest mentioning that 26 systematic reviews met all eligibility criteria (similar to lines 213-215).

4) In the last sentence of the Abstract Methods and Findings section, please describe the main limitation(s) of the study's methodology.

AUTHOR SUMMARY

In the final bullet point of 'What Do These Findings Mean?', please include the main limitations of the study in non-technical language. 

INTRODUCTION

1) l Please cite the reference numbers in square brackets. Citations should precede punctuation. Please revise throughout.

2) l.104: Please define ‘US’ at first use.

3) l.120: Please define ‘UK’ at first use.

4) l.123: Please exchange the word ‘elderly’ with ‘older’.

5) l.127: Please define ‘BMI’ at first use.

6) l.128: The terms gender and sex are not interchangeable (as discussed in https://www.who.int/health-topics/gender#tab=tab_1); please use the appropriate term.

METHODS AND RESULTS

1) l.163: Please specify “adult patients”, e.g., by providing an age range.

2) l.243: "(6, 55%)" - we suggest to consistently report the data as (6/11, 55%), i.e. numerator/denominator and percentage. Please revise throughout.

3) l.249: Please define ‘JBI’ and ‘CASP’ first use.

4) Table 2: Please consider avoiding the use of red and green in order to make your figure more accessible to those with colour blindness

5) l.302: Please define ‘CI’ at first use. Throughout, we suggest reporting statistical information as follows to improve clarity for the reader "22% (95% CI [13%,28%]; p</=)". When reporting 95% CIs please separate upper and lower bounds with commas instead of hyphens as the latter can be confused with reporting of negative values.

6) Table 3: Please define ‘PROBAST’ below the table.

7) Table 4: Please define ‘PI’ below the table.

8) Table 5: Please check whether the abbreviations ‘CASP’ and ‘S.S.’ appear in the table and remove accordingly.

DISCUSSION

1) Please remove any subheadings from the discussion section, including the ‘conclusion’ subheading.

2) l.571: Please define ‘DTA’ at first use.

REFERENCES

1) PLOS uses the numbered citation (citation-sequence) method and first six authors, et al.

2) Please ensure that journal name abbreviations match those found in the National Center for Biotechnology Information (NCBI) databases (http://www.ncbi.nlm.nih.gov/nlmcatalog/journals), and are appropriately formatted and capitalised.

3) Where website addresses are cited, please include the complete URL and specify the date of access (e.g. [accessed: 12/06/2024]).

4) Please also see https://journals.plos.org/plosmedicine/s/submission-guidelines#loc-references for further details on reference formatting.

SUPPLEMENTARY MATERIAL

1) Please note that supplementary material will be posted as supplied by the authors.

2) In the published article, supporting information files are accessed only through a hyperlink attached to the captions. For this reason, you must list captions at the end of your manuscript file. You may include a caption within the supporting information file itself, as long as that caption is also provided in the manuscript file. Do not submit a separate caption file.

When SI files are contained with a single file:

Please label the file as ‘S1 Supporting Information’.

Please apply alphabetical labelling to each table and figure contained within the S1 file. For example, ‘Fig A’ to ‘Fig Z’ and ‘Table A’ to ‘Table Z’.

Plain text does not need to be labelled and can just be given a title as necessary. For example, ‘Statistical Analysis Plan’.

Please cite tables/figures as ‘Fig A in S1 Supporting Information’ and/or ‘Table A in S1 Supporting Information’, for example.

Please cite plain text as, ‘Statistical Analysis Plan in S1 Supporting Information’, for example.

When SI files are uploaded as separate files:

Please label tables as ‘S1 Table’ (so on) and figures as ‘S1 Fig’ (and so on).

Any additional documents (protocols/analysis plans etc.) can be labelled as ‘S1 Protocol’, for example. Please cite items as exactly as labelled.

SOCIAL MEDIA

To help us extend the reach of your research, please provide any X (formerly known as Twitter) handle(s) that would be appropriate to tag, including your own, your co-authors’, your institution, funder, or lab. Please enter in the submission form any handles you wish to be included when we post about this paper.

General Editorial Requests

---

## [Editor Report · Decision Letter 3]

20 Dec 2024

Dear Dr Dinnes, 

On behalf of my colleagues and the Guest Academic Editor, Janet Cuddigan, I am pleased to inform you that we have agreed to publish your manuscript "Accuracy and clinical effectiveness of risk prediction tools for pressure injury occurrence: An umbrella review" (PMEDICINE-D-24-00795R3) in PLOS Medicine.

I appreciate your thorough responses to the reviewers' and editors' comments throughout the editorial process. We look forward to publishing your manuscript, and editorially there is only one remaining minor stylistic point that should be addressed prior to publication. We will carefully check whether the change has been made. If you have any questions or concerns regarding these final requests, please feel free to contact me at atosun@plos.org.

Please see below the minor point that we request you respond to:

1) References: Where website addresses are cited, please specify the date of access by providing day, month and year (e.g. [accessed: 12/06/2024]).

Before your manuscript can be formally accepted you will need to complete some formatting changes, which you will receive in a follow up email (including the editorial point above). Please be aware that it may take several days for you to receive this email; during this time no action is required by you. Once you have received these formatting requests, please note that your manuscript will not be scheduled for publication until you have made the required changes.

PRESS

Sincerely, 

Alexandra Tosun, PhD 

Associate Editor 

PLOS Medicine